

# The impacts of secondary ice production on microphysics and dynamics in tropical convection

Zhipeng Qu[1], Alexei Korolev[1], Jason A. Milbrandt[2], Ivan Heckman[1], Yongjie Huang[3], Greg M. McFarquhar[4,5], Hugh Morrison[6], Mengistu Wolde[7], Cuong Nguyen[7]

[1]Meteorological Research Division, Environment and Climate Change Canada, Toronto, Ontario, Canada
[2]Meteorological Research Division, Environment and Climate Change Canada, Dorval, Quebec, Canada
[3]Center for Analysis and Prediction of Storms, University of Oklahoma, Norman, OK, USA
[4]Cooperative Institute for Severe and High Impact Weather Research and Operations, University of Oklahoma, Norman, OK, USA
[5]School of Meteorology, University of Oklahoma, Norman, OK, USA
[6]Mesoscale and Microscale Meteorology Laboratory, National Center for Atmospheric Research, Boulder, CO, USA
[7]National Research Council Canada, Ottawa, Canada

*Correspondence to*: Zhipeng Qu (zhipeng.qu@ec.gc.ca)

**Abstract.** Secondary ice production (SIP) is an important physical phenomenon that results in an increase of ice particle
concentration and can therefore have a significant impact on the evolution of clouds. In this study, idealized simulations of a mesoscale convective systems (MCS) was conducted using a high-resolution (250-m horizontal grid spacing) mesoscale model and a detailed bulk microphysics scheme in order to examine the impacts of SIP on the microphysics and dynamics of a simulated tropical MCS. The simulations were compared to airborne *in situ* and remote sensing observations collected during the High Altitude Ice Crystals - High Ice Water Content (HAIC-HIWC) field campaign in 2015. It was found that simulated
ice particle size distributions are highly sensitive to the parameterization of SIP. Inclusion of SIP processes in the microphysics scheme is crucial for the production and maintenance of high ice water content in the simulated tropical convection. It was shown that SIP can enhance the strength of the existing convective updrafts and result in the initiation of new updrafts above the melting layer. Agreement between the simulations and observations highlights the impacts of SIP on the maintenance of tropical MCSs in nature and the importance of including SIP parameterizations in models.

## 1 Introduction

Secondary ice production (SIP) is recognized as a fundamental cloud microphysical process (e.g., Cantrell and Heymsfield, 2005; Field et al. 2017). Production of secondary ice involves processes that require the presence of pre-existing ice particles. SIP is different from primary ice production (PIP), which commences by the nucleation of ice either homogeneously in strongly supercooled droplets or heterogeneously on the surface of ice-nucleating particles (INPs) (e.g., Kanji et al., 2017).

The first *in situ* observations of SIP go back to the early 1960s (e.g., Murgatroyd and Garrod, 1960; Koenig, 1963, 1965). Multi-year *in situ* measurements have shown that SIP is an very common phenomenon, and it occurs in different types of





clouds from polar regions to the tropics (recent SIP studies, e.g., Lloyd et al., 2015; Lawson et al., 2015, 2017; Lasher-Trapp et al., 2016; Keppas et al.,2017; Mignani et al. 2019; Korolev et al., 2020; Li et al., 2021; Luke et al. 2021 and many others).

The primary effect of SIP is the enhancement of ice particle concentration which, depending on environmental conditions,

may exceed the concentration of PIP ice particles by several orders of magnitude (e.g., Hobbs and Rangno, 1985; Ladino et al., 2017). Such an enhancement of ice particle concentration may have a significant effect on the phase composition, cloud dynamics, precipitation rate, and cloud radiative properties, impacting the energy balance and hydrological cycle on regional and global scales.

At present, six mechanisms are recognized as sources of secondary ice in clouds. These include the fragmentation of

freezing droplets (hereafter FFD) (e.g., Kleinheins et al. 2021), rime splintering (i.e., the Hallett-Mossop process, hereafter HM) (e.g., Hallett and Mossop, 1974), fragmentation due to ice-ice collisions (e.g., Vardiman 1978; Takahashi et al. 1995), ice fragmentation due to thermal shock (e.g., Dye and Hobbs, 1968), fragmentation of sublimating ice (Oraltay and Hallett, 1989), and activation of INPs in transient supersaturation around freezing drops (e.g., Prabhakaran et al., 2020). A detailed description of these SIP mechanisms and the status of associated laboratory studies are discussed in the review of Korolev and

Leisner (2020). It was found that HM and FFD are the most experimentally studied SIP mechanisms, and in which production rates of secondary ice have been quantified. However, a detailed analysis of previous experiments by Korolev and Leisner (2020) revealed a large diversity of the ice production rates, which led to the conclusion that these SIP processes need to be studied further. The other four mechanisms have a limited number of laboratory experiments, and cover only a fraction of environmental conditions (e.g., fragmentation during ice collisions, fragmentation of sublimating ice), or only demonstrated

the general feasibility of SIP mechanisms (e.g., fragmentation due to thermal shock, activation of INPs in transient supersaturation around freezing drops). All these led Korolev and Leisner (2020) to the conclusion that the relative contributions of each of the six SIP mechanisms in the enhancement of ice concentrations remain uncertain.

Despite its apparent geophysical significance, systematic studies of the effect of SIP on cloud microphysics with the help of cloud simulations have begun only in the last few years (e.g., Phillips et al. 2017a, 2018; Sullivan et al. 2018; Hoarau et al.

2018; Fu et al. 2019; Sotiropoulou et al. 2020, 2021; Dedekind et al. 2021; Hawker et al. 2021; Huang et al., 2021, 2022 and others). Most of these modeling efforts were focused on matching simulated moments of particle size distributions (PSDs) with those observed *in situ*. In many ways, the implementation of SIP in numerical models was hindered by the lack of consensus on parameterizations of SIP mechanisms.

One of the main objectives of this work is to identify and simulate the occurrence of high ice water content (IWC) associated

with the enhancement of ice particle concentrations from SIP processes. Cloud environments with high IWC ($> 1$ g m$^{-3}$) pose a hazard for civil aviation and may result in engine power loss, stall, or damages (e.g., Lawson et al. 1998; Mason et al. 2006, Mason and Grzych, 2011). The phenomenon of high IWC is well documented from *in situ* observations in tropical mesoscale convective systems (MCSs) (e.g., Heymsfield and Palmer 1986, Lawson et al. 1998, Gayet et al. 2012, Fridlind et al. 2015; Leroy et al. 2017, Strapp et al. 2021). Several previous modeling studies using different cloud microphysical parameterizations

attempted to reproduce high IWCs. Ackerman et al. (2015) used a 1D model to explore microphysics in tropical MCSs.



Simulations performed with 3D models (Franklin et al. 2016; Stanford et al. 2017; Qu et al. 2018) pointed to the inaccuracies in the estimation of cloud PSD, IWC, and ice category comparing to the observations. Huang et al. (2021) conducted high-resolution simulations of tropical convection and found significant overestimates of radar reflectivity and underestimates of total ice crystal concentration (Ni). Adding SIP in high-resolution simulations, Huang et al. (2022) found significant

improvement of simulated $N_i$ compared to the *in situ* observations.

In most of the previous numerical studies investigating SIP, the microphysics schemes used were based on the traditional approach of representing ice-phase hydrometeors whereby they are partitioned into various predefined categories (e.g., pristine ice, snow, graupel, etc.) with prescribed physical properties. This approach has several inherent limitations and problems, including a limited range of ice properties (e.g., bulk density) that can be represented, inconsistent physical processes applied

to the categories, and the need to parameterize conversion between categories, an artificial process which can not be constrained from observations and is purely *ad hoc*. To address this problem, Morrison and Milbrandt (2015) proposed a new approach and developed a new microphysics parameterization – the Predicted Particle Properties (P3) scheme –whereby all ice-phase hydrometeors are represented by a single "free" ice category whose physical properties evolve continuously. While flexible in this regard, one limitation of the original P3 scheme was that it could not represent more than one population of ice

particles (with different bulk properties) at a given time and grid location. The scheme was thus generalized  to allow for a user-specified number of "free" ice categories, each of which have properties that evolve continuously and can represent any ice type (Milbrandt and Morrison 2016).

The P3 scheme was used in the tropical convection simulations of Huang et al. (2022). While they found significant improvement of simulated $N_i$ compared to the *in situ* observations by adding three SIP mechanisms, their simulations were

limited to two ice categories. This is the minimum number of categories required for including SIP processes, since at least two categories are needed to represent the co-existence of newly formed small ice splinters and pre-existing large ice particles. However, as will be shown below, the use of more than two ice categories may be beneficial or even necessary to model the impacts of SIP in deep convection.

This study is focused on the examination of the effects of SIP on the microphysics and dynamics of a simulated tropical

MCS. Quasi-idealized simulations of a MCS were conducted using a near cloud-resolving configuration (250-m horizontal grid spacing) of a 3D dynamical model with the P3 microphysics scheme. Model configurations with up to four free ice categories were tested. The enhancement of ice particle concentration by SIP is represented by HM and FFD mechanisms. In the absence of a consensus on SIP parameterizations, these two processes were described by simplified parameterizations, which provide a sufficient enhancement of $N_i$ above the melting layer consistent with *in situ* observations in the MCSs (Korolev

et al. 2020). Simulated ice PSD, $N_i$, IWC, radar reflectivity, and Doppler velocity were compared against *in situ* and remote sensing observations collected during the HAIC-HIWC field campaign in 2015 (Strapp et al. 2021). Without looking for exact match between model simulations and observations, this study aimed to show whether the simulation with SIP produces better estimation of the observed microphysics compared to the simulation without SIP.



The remainder of the paper is structured as follows. The next section describes the observation data used for evaluation. In
section 3, the setup of the model, the microphysics scheme, and the parameterizations of SIP are detailed. Section 4 describes
the choice of control simulation with regard to the number of ice categories in P3. Section 5 assesses the impact of SIP on the
formation of ice clouds based on the control simulation. The role of SIP in strengthening and sustaining tropical convection is
discussed in section 6. This is followed by an assessment of the impact of ice-ice collection efficiency on the simulation. The
final section offers a summary of the study and conclusions.

## 2. Observation data

*In situ* data employed in this study were collected from the National Research Council of Canada (NRC) Convair-580 and
Service des Avions Français Instrumentés pour la Recherche en Environnement (SAFIRE) Falcon-20 research aircrafts. The
coordinated flight operations of the NRC Convair-580 and SAFIRE Falcon-20 in the frame of the HAIC-HIWC campaign
were performed out of Cayenne (French Guiana) during May 2015.

The measurements of PSDs were performed by three particle probes, which covered different particle size ranges. The
Droplet Measurement Technologies (DMT) Cloud Droplet Probe (CDP: Lance et al. 2010) was used for measurements of
droplets in size range $2\ \mu m < D < 50\ \mu m$. The Stratton Park Engineering Company (SPEC) 2D imaging-stereo (2D-S: Lawson
et al. 2006) covered the nominal size range from 10 to 1250 $\mu m$. The DMT Precipitation Imaging Probe (PIP: Baumgardner
et al. 2001) provided measurements of particles in the nominal size range from 100 $\mu m$ to 6.4 mm (PSD). The processing
software employed a retrieval algorithm of partially viewed particle images (Heymsfield and Parrish 1979; Korolev and
Sussman 2000), which allowed the enhancement of particle statistics and extended the maximum size of the composite PSD
up to 12.8 mm.

All particle probes were equipped with anti-shattering tips to mitigate the effect of ice shattering on the measurements of
ice particle concentration. Residual shattering artifacts were identified and filtered out with the help of the inter-arrival time
algorithm (Field et al. 2006; Korolev and Field 2015).

The bulk IWC was measured by an Isokinetic Probe (IKP: Davison et al. 2008). The IKP allowed measurements of IWC
up to 10 g m$^{-3}$ at the aircraft speed 200 m s$^{-1}$. Such a high upper limit of IWC well exceeded maximum IWC (~5 g m$^{-3}$)
measured during the HAIC-HIWC campaign and ensured that the measured IWC never exceeded the IKP saturation level.

Both aircraft were equipped with the same instruments for measurements of PSDs and bulk IWC and used the same
processing algorithms applied to these measurements. Such arrangement minimized differences in systematic errors specific
to different types of instruments and synchronized data processing. Therefore, if any potential biases in data acquisition and
data processing existed, they would be the same for both data sets collected from the NRC Convair-580 and SAFIRE Falcon-
20.

Besides comparisons with IWC and $N_i$, model results are also compared with reflectivity and Doppler velocity measured
by the NRC aircraft X-band radar (NAX) installed on the NRC Convair-580 (Wolde and Pazmany 2005). Statistics of the



NAX data included the MCS cloud segments that precipitated down to the ground surface level. Cloud segments with outflow cirrus having radar returns disconnected from the ground were excluded from the statistics.

## 3. Model configuration

### 3.1. Atmospheric model and initialization

The model used in this study is the Global Environmental Multiscale (GEM) model (Côté et al. 1998; Girard et al. 2014). GEM is used for operational numerical weather prediction (NWP) in Environment and Climate Change Canada (ECCC) as well as research in ECCC and Canadian universities. The dynamical core of GEM is formulated based on the non-hydrostatic fully-compressible primitive equations with a terrain-following hybrid vertical grid. As such, it can be run at cloud-resolving (sub-km grid spacing) scales. It can be run on global or limited-area domains and is capable of one-way nesting. In this study,

an idealized model configuration was used to simulate tropical deep convection, with a horizontal grid spacing of 250 m in a simulation domain of 160 km × 160 km, with 83 vertical levels over a tropical ocean surface. The horizontal grid spacing of 250 m is nearly at the cloud-resolving scale (Bryan et al. 2003; Lebo and Morrison, 2015). It is also close to the corresponding distances (110 to 180 m) of the 1 Hz *in situ* observation data from the aircraft which flew mostly at 110-120 m s$^{-1}$ for Convair-580 and 150-180 m s$^{-1}$ for Falcon-20. To resolve the vertical profiles near the tropical melting layer, vertical grid spacings of

approximately 100 m were used between the altitudes of 4 and 7.5 km. The readers are referred to Table 1 for more details of dynamics/numerics and physics configurations used in the model.

    The atmospheric initial conditions were horizontally homogeneous, based on an initial sounding taken from the operational global GEM analysis at 12 UTC on 15 May 2015 at 6.769° N and 49.551° W. The initial profile (Fig. 1) had 1697 J kg$^{-1}$ of convective available potential energy (CAPE). The GEM analysis also provided the initial sea surface temperature. The

location and time were chosen based on the occurrence of an extensive mesoscale system that formed in this region and was observed (Fig. 2) during the HAIC – HIWC field campaign (Strapp et al. 2021).

    To initiate the model storm, the updraft nudging method of Naylor and Gilmore (2012) was used to force convection during the first 15 min of the simulation. Three distinct updrafts 15 km apart from each other in the western part of the simulation domain were initialized. Each updraft was forced by perturbing the vertical air velocity ($w_t$) in a spheroid with a horizontal

radius of 10 km and vertical radius of 1.5 km, centered at 1.5 km altitude:

$$w_{mag} = \begin{cases} w_{max} cos^2 \left( \frac{\pi}{2} \beta \right), & if\ 0 \leq \beta \leq 1, \\ 0, & if\ \beta > 1 \end{cases} \tag{1}$$

$$w_t = w_{t-1} + dts \times \alpha \times \max(w_{mag} - w_{t-1}, 0), \tag{2}$$

    where $\beta$ is the distance from the center of the spheroid normalized by its radius, α is an inverse nudging timescale (0.5 s$^{-1}$), $dts$ is the model time step, $w_{max}$ is the maximum updraft speed (10 m s$^{-1}$) for nudging.



## 3.2. Cloud microphysics scheme


All cloud microphysical processes in the GEM simulations were represented by the P3 two-moment bulk microphysics scheme, where up to four (free) ice categories were used. For each ice category, there are four prognostic (i.e. advected) mixing ratio variables: the total ice mass, the rime mass, the bulk volume, and the total number. From the prognostic variable fields, various bulk physical properties can be computed. The size distribution of each category is represented by a complete gamma

function. The liquid-phase component of P3 consists of two-moment categories for cloud droplets and rain. For details on the representation of each hydrometeor category and the parameterized microphysical processes, readers are referred to Morrison and Milbrandt (2015); for details on the multi-category configuration, see Milbrandt and Morrison (2016).

It should be noted that there have several further key developments to P3 since the first version of the multi-ice-category scheme, along with various minor modifications. Major developments include a triple-moment treatment of rain (Paukert et

al. 2019), introduction of a prognostic liquid fraction for wet ice (Cholette et al. 2019), and a triple-moment treatment of ice (Milbrandt et al. 2021). The version of P3 used in this study does not include these major modifications. The impacts of these components of P3 on SIP may be examined in future work. It is possible, for example, that triple-moment ice, which in principle results in a better representation of the PSD dispersion, may be important for some aspects of modeling SIP and its impacts. However, such work is outside of the scope of this study.

## 3.3. Parameterization of SIP


The following two SIP mechanisms were examined in this study. Other mechanisms will be considered in the future.

### 3.3.1 Rime splintering/Hallet-Mossp (HM)

The pristine version of P3 used in this study (i.e., prior to SIP-related modifications examined here) includes a parameterization of the HM mechanism if two or more ice categories are used. The requirement for at least two categories is

to prevent dilution of ice particle properties when two populations of ice are forced to be represented by a single size distribution and set of physical properties (see Milbrandt and Morrison 2016). The parameterized HM process produces a maximum of 350 ice crystals per mg of collected liquid water, with crystal sizes of 10 µm, during riming of rain within a temperature range of $-3°C > T > -8°C$, with the peak value at $-5°C$ and varying linearly to 0 at the extreme temperature ranges. This ice multiplication parameterization has been used in several traditional (fixed ice category) microphysics schemes (e.g.,

Reisner et al. 1998). This is similar to the original HM parameterization in P3 as used in Milbrandt and Morrison (2016).

One modification for HM parameterization in this study is to exclude the use of collected liquid water from the situation where raindrops (100 µm $< D <$ 3500 µm) were collected/nucleated by small ice particles ($D <$ 100 µm). From the point of view of SIP mechanism, it is more appropriate to apply this part of collected liquid water to the FFD mechanism.



### 3.3.2 Fragmentation of freezing drops (FFD)

The parameterization of the FFD mechanism was implemented following Lawson et al. (2015):

$$N_f = 2.5 \times 10^{-11} D^4 \tag{3}$$

where $N_f$ is the average number of ice fragments per drop, and $D$ is the drop diameter in micrometers. The parameterization of the FFD process was applied for raindrops (100 µm < $D$ < 3500 µm) which were nucleated by ice particles ($D$ < 100 µm). Following Keinert et al. (2020) the activity of the FFD process was limited to the temperature range -25°C < $T$ < -2°C.

### 3.3.3. Ice collection efficiency


Although not directly part of SIP, ice aggregation is another process that impacts $N_i$ and may therefore be important in affecting high IWC. There are several different approaches to parameterize the ice collection efficiency, which is a key parameter in the aggregation parameterization (Hallgren and Holster 1960; Lin et al. 1983; Cotton et al. 1986; Ferrier et al. 1994, 1995; Milbrandt and Yau 2005a,b; Seifert and Beheng 2006). Khain and Pinsky (2018) showed that the collection

efficiencies in these parameterizations vary by more than two orders of magnitude for any given temperature between -40°C and 0°C. Sensitivity simulations conducted in the frame of this current work showed a high sensitivity of the modeled IWC and $N_i$ to the collection efficiency of ice. The collection efficiency used in this study follows Cotton et al. (1986). Further discussion of the sensitivity of the modeling results to the ice aggregation parameterization is presented in section 7.

### 4. Establishment of the control configuration

The number of ice categories used in the P3 scheme can impact the overall simulation results (Milbrandt and Morrison, 2016). SIP results in large quantities of small ice splinters, which can be co-located with pre-existing larger ice particles, and thus bimodal or multi-modal ice size distributions may occur. This cannot be represented with the single-ice-category configuration since P3 uses complete gamma size distributions for each hydrometeor category. Thus, SIP – or any other ice initiation process – could result in the "dilution" of the bulk particle properties of existing ice where the microphysics scheme

tries to represent two or more populations of ice particles within a single size distribution and with a single set of bulk physical properties (e.g., mean size). Thus, before examining the impacts of SIP in the simulation, a control configuration must be established based on the minimum number of ice categories needed to represent ice particle evolution in P3 with sufficient detail. This is determined by the number of categories beyond which adding more does not change the simulated average profiles for more than 15% for the fields of interest (e.g., IWC and $N_i$). To address this a series of sensitivity tests was conducted

using the baseline version of P3 (with no SIP), varying the number of ice categories from 1 to 4, and using P3 with SIP included, varying the number of categories from 2 to 4.

Table 2 summarizes the complete list of experiments conducted in this study. For the experiments starting with "BASE" (denoting the baseline P3 configuration), no SIP is activated. The experiments starting with "SIP" use the new parameterization



for both FFD and HM process. In the following analysis, we focus on the simulation results between 90 and 150 min, when
the convective systems are more complex and more closely resemble the observed MCS.

As described in section 3, three convective cells were initiated in the eastern part of the simulation domain. Figure 3 shows
the upward longwave flux from the BASE-1ICE simulation at the top of the atmosphere (Figs. 3a-d) and the radar reflectivity
of the vertical cross-sections (Figs. 3e-h) indicated by black lines in (Figs. 3a-d) for simulation times 30, 60, 120, and 180 min.
The initial formation of the three convective updrafts can still be seen at 30 min (Fig. 3a). By 60 min, these updrafts started to
merge, forming a larger system (Fig. 3b). This system then moved westward (towards the right of the domain) and developed
into a sustained system (Fig. 3c). By 180 min the convection began to weaken (Figs. 3d, h).

Figure 4 shows averaged profiles for the baseline simulations with different numbers of ice categories. The mean profiles
of IWC (Fig. 4a) and $N_i$ (Fig. 4b) consider all points with IWC larger than 0.001 g m$^{-3}$. The maximum vertical wind speed
($w_{max}$; Fig. 4c) and average temperature profiles (Fig. 4d) apply to the entire model domain. The mean rainwater profiles (Fig.
4e) are calculated for the area with both ice water path and rain water path larger than 1 g m$^{-2}$. The radar reflectivity profiles
(Fig. 4f) are the median values including points with either IWC or rain water content (RWC) larger than 0.01 g m$^{-3}$ within the
mask used for Fig. 4e.

For the altitude range between 5 and 12 km, adding one more ice category to one, two and three-category baseline
simulation produce maximal changes of 12%, 23% and 7% for IWC respectively (Fig. 4a). Similarly, the maximal changes of
25%, 31% and 14% are found for $N_i$ (Fig. 4b). The radar reflectivity (Fig. 4f) for the one-ice category run (BASE-1ICE) is
about 4 to 6 dBZ lower than the three or four-category simulations in the same altitude range. This is likely caused by the fact
that a single ice category is not sufficient to represent the co-existence of large and small ice particles and results in reduction
of the concentration of large ice particles, leading to lower radar reflectivity. With regard to the number of ice categories for
SIP simulations, a similar conclusion to the baseline simulations was found. Adding one more ice category to two and three-
category SIP simulations produce maximal changes of 31% and 9% for IWC respectively (Fig 5a). For $Ni$, the maximal changes
of 70% and 15% are found (Fig 5b). Therefore, at least three ice categories in P3 appear to be necessary and sufficient to
examine the impacts of including SIP processes. It is of passing interest to note that the similarity of the three and four-ice
category results is consistent with the 1D kinematic simulations in Milbrandt and Morrison (2016). However, given that the
use of more ice categories in P3 is generally preferable in principle (though there is added computational cost), and that the
four-ice-category simulations were already performed, BASE-4ICE is taken as the control run for the sensitivity studies to
follow; this simulation is referred to as CTR. Correspondingly, we focus on the four ice category simulation including SIP
(SIP-4ICE) for direct comparison to CTR.



## 5. Impacts of SIP on microphysical properties

### 5.1. Domain-averaged profiles

Figure 6 shows simulated average profiles for the four-ice category simulation including SIP processes, SIP-4ICE (see Table 2) as well as for CTR. As seen in Fig. 6a, the SIP-4ICE simulation has at least 100% higher IWC compared to the control run above 6 km. SIP-4ICE has significantly higher $N_i$ than CTR (Fig. 6b), with differences reaching two to three orders of magnitude near 6 km and about one order of magnitude above 11 km.

Figure 6c shows the maximum vertical air velocity in the domain. The simulations are very similar below the melting layer

(~4.5 km), however, above the melting layer, $w_{max}$ of SIP is 2 to 5 m s$^{-1}$ higher. This suggests that SIP enhances convection due to the sudden production of a large number of small ice particles resulting in the rapid freezing of rainwater and depletion of water vapor by diffusional growth. Both effects result in latent heating, which invigorates the convection. This can be inferred from Fig. 6e showing that RWC in CTR is reduced form 0.12 g m$^{-3}$ at 4.2 km to 0.05 g m$^{-3}$ at 5 km, whereas RWC form SIP-4ICE is changed from 0.12 g m$^{-3}$ at 4.2 km to 0.01 g m$^{-3}$ at 5 km. Another potential mechanism of convection

enhancement above the melting layer will be discussed in section 6.

The medians of radar reflectivity of the SIP simulation is 5 to 10 dBZ lower compared to those of the CTR between the altitudes of 5 and 12 km (Fig. 6f). This is because the SIP simulation has smaller ice particles, despite the higher IWC values, due to the higher $N_i$.

In order to confirm that the simulation differences illustrated in Fig. 6 are indeed the result of the inclusion of parameterized

SIP processes, and not simply a chance set of changes that could result from perturbations in the model due to some minor code change, a set of 10 ensemble simulations were run for each configuration (CTR and SIP-4ICE) but with perturbed initial conditions. Figure 7 shows the ensemble results for the two configurations at 120 min. Each configuration includes 11 members (1 unperturbed + 10 perturbed members). The initial temperature profile is randomly perturbed with the maximum range of ±1°C at all model levels. The ensemble results show high consistency. The SIP simulation consistently produces much higher

IWC and $N_i$. The vertical velocity $w_{max}$ diverges above the altitude of ~6 km, which indicates that the SIP simulations generate stronger updrafts in general. The RWC and the radar reflectivity are both strongly reduced by 5 to 10 dBZ in the SIP simulations between 5 and 10 km. Similar remarks can be made for all other times between 90 and 150 min (not shown). The results summarized in Fig. 7 lend support to the idea that the differences between CTR and SIP-4ICE are indeed due to the effects of SIP on the microphysical and thermodynamic fields, and not merely another model realization of a chaotic weather system.

### 5.2. Ice number concentration

Figure 8 shows comparisons of the probability density function of $N_i$, $F(N_i)$, calculated from the simulations and measured during airborne *in situ* observations at two different altitude ($H$) ranges. The $N_i$ measured *in situ* is calculated for size range between 40 μm and 12.5 mm. The ice particles smaller than 40 μm is not counted due to large uncertainty of the instrument for the size range (Baumgardner et al. 2017). The $F(N_i)$ measured *in situ* were averaged over all HAIC-HIWC flights for the





clouds with IWC > 0.01 g m$^{-3}$. Figure 8a shows a comparison of $F(N_i)$ at altitudes $6 < H < 7$ km. The CTR simulation (blue line) shows a significant underestimation of $N_i$ compared to the measured values (black line). The concentrations corresponding to the modal value of $F(N_i)$ in CTR are nearly two orders of magnitude lower than those obtained from *in situ* observations. The SIP-4ICE simulation (red line) shows good agreement with the measured values.

The general behaviour of the functions $F(N_i)$ for $11 < H < 12$ km (Fig. 8b) is similar to that obtained for $6 < H < 7$ km in
Fig.8a. The maxima of $F(N_i)$ for CTR and observed values correspond to approximately the same ice concentrations of $10^5$ m$^{-3}$. However, the maximum of $F(N_i)$ in CTR is nearly triple that of the observed $F(N_i)$ which was reasonably close to that of SIP-4ICE. There are almost no grid points in CTR with a concentration above $\sim 3 \times 10^5$ m$^{-3}$, whereas SIP-4ICE overestimated $F(N_i)$ compared to the measured values.

**5.3 Ice water content**

Similar to the results discussed above, Fig. 9 shows comparison of probability density functions of IWC, $F(IWC)$, obtained from model simulations and aircraft observations at altitudes $6 < H < 7$ km (Fig. 9a) and $11 < H < 12$ km (Fig. 9b). As seen in Fig. 9a, $F(IWC)$ from SIP-4ICE is in good agreement with the observations for IWC smaller than 3.25 g m$^{-3}$. In contrast, the simulated frequency of encountering high IWC in CTR is about ½ to 1/500 of the observed frequency between IWC of 1 and 2 g m$^{-3}$. There is no data with IWC higher than 2.5 g m$^{-3}$ in CTR.

SIP-4ICE produces some points with IWC > 3.25 g m$^{-3}$, which were not observed by the instruments. The $F(IWC)$ of these high IWC conditions from SIP-4ICE are below $1.7 \times 10^{-5}$. For the Convair-580 aircraft, the number of observed 1-s average data points with IWC > 0.01 g m$^{-3}$ is 59,893. This sets the limit of $F(IWC)$ of the observation data at $1.7 \times 10^{-5}$. If the campaign lasted much longer, it is conceivable that these high IWC conditions might have eventually been observed.

Figure 9b shows similar results but for higher altitudes (between 11 and 12 km). The frequency of IWC between 0.3 and
1.3 g m$^{-3}$ in CTR is about 2 to 3 orders of magnitude lower than the observed frequency. There is no data with IWC higher than 1.30 g m$^{-3}$ from CTR. SIP-4ICE produces closer estimates compared to the observation data as they both have IWC up to $\sim 3$ g m$^{-3}$. Between 11 and 12 km, $N_i$ of SIP-4ICE is considerably improved compared to CTR as shown in Fig. 8b. However, the IWC of SIP-4ICE at these altitudes is still underestimated by 1 to 2 orders of magnitude beyond IWC of 0.7 g m$^{-3}$ compared to the observation. One possible reason for this underestimate is the differences in the sampling of data. At higher altitudes
between 11 and 12 km, there are often extensive areas with thin ice clouds near the convection. The data from SIP-4ICE used in the statistics include these areas if the IWC of the grid cell larger than 0.01 g m$^{-3}$. In contrast, the HAIC-HIWC campaign targeted conditions with high IWC. The thinner ice clouds with IWC between 0.01 and 0.3 g m$^{-3}$ might not have been sufficiently sampled as they are less relevant to the extreme conditions causing safety issues for aviation. This difference might partly explain the higher $F(IWC)$ below 0.3 g m$^{-3}$ and lower $F(IWC)$ above 0.3 g m$^{-3}$ for SIP-4ICE compared to the
observations.



### 5.4. Longwave radiation and radar reflectivity

Figure 10 shows the upward longwave radiative flux at the top of atmosphere (TOA) for three different simulation times (90, 120, and 150 min) from CTR (Figs. 10a-c) and SIP-4ICE (Figs. 10e-f). The lowest TOA flux of SIP-4ICE is on average 11.4°C lower than that of CTR. The surface of area with TOA longwave flux lower than 170 W m$^2$ of SIP-4ICE is on average
2.7 times larger than that of CTR. These suggest that the cloud tops with SIP included are higher (Figs. 10d, e) and the anvil clouds are more extensive (Figs.10e, f).

The corresponding simulated radar reflectivity of the cross-section indicated by black lines in Fig. 10 is shown in Fig. 11. One significant difference between the CTR (Figs. 10a-c) and SIP-4ICE simulation (Figs. 10d-f) is that the reflectivity from the simulation with SIP is significantly lower than that of the control run between altitudes of approximately 5 and 10 km.
This is due to the higher $N_i$ values and thus smaller ice particle sizes in SIP-4ICE.

Figure 12 shows a comparison of the frequency distribution of radar reflectivity for CTR and SIP-4ICE (Figs. 12a, b) and for the NRC Convair-580 X-band radar (Fig. 12c). For the results of the two simulations, only the atmospheric columns with both ice water path and rain water path larger than 1 g m$^{-2}$ are selected. The X-band radar data in Fig. 12c was averaged over all research flights during the HAIC-HIWC campaign. Figure 12d shows the simulated and observed median values of the
reflectivity. At altitudes higher than 10.3 km reflectivity in both the CTR and SIP-4ICE runs is lower than the measured reflectivity. However, between 5 and 10 km, SIP-4ICE has values closer to the observations with a maximum overestimation of 4 dBZ at 5 km. CTR clearly overestimates the reflectivity by 5 to 15 dBZ between 5 and 10 km.

Note that at the altitude of the melting layer (~4.5 km) none of the simulations reproduce the distinct bright band that is clearly apparent in the observation data. This is due to the fact that the version of P3 used in this study does not properly
represent the transition state of melting ice with a wet surface and ice core (nor does P3 artificially boost the reflectivity contribution from ice during melting in order to mimic a bright-band effect). As mentioned in section 3.3, a newer version of P3 includes a prognostic variable for the liquid mass content for each ice-phase category, which allows for mixed-phase particles and a corresponding improvement in the calculation of reflectivity in the melting zone (to be shown in a forthcoming publication).

**5.5. Vertical Doppler velocity**

The Doppler velocity from the simulation is calculated using mass weighted fall speed of all hydrometeors subtracted from the vertical wind speed. Figure 13 shows the simulated Doppler velocity for CTR and SIP-4ICE for the vertical cross section shown in Fig. 11. The Doppler velocity below the melting layer is mostly negative due to the high fall speed of the rain. Above the melting layer the average Doppler velocity gradually increases with altitude from -2 to 0 m s$^{-1}$ between 5 and 14 km. The
gradual increase of the Doppler velocity is primarily linked to the size of ice particles. Positive values of the Doppler velocity are associated with convective cloud regions where the updraft velocity exceeds the falls speed of the ice particles. One



difference between the control and SIP simulations is that the SIP-4ICE simulation has higher Doppler velocity between 5 and 8 km for all three times analyzed.

Figure 14 shows similar results to Fig. 12 but for the Doppler velocity. To mitigate the effect of anvils the diagrams in

Figs. 14a and b used the same mask as that for radar reflectivity in Figs. 12a and b. Both CTL (Fig. 14a) and SIP-4ICE (Fig. 14b) show distribution patterns very similar to those obtained from the observations (Fig. 14c). Figure 14d shows comparisons of the simulated and observed median values of the Doppler velocity versus altitude. The SIP-4ICE simulation produces very close results to the measurements between the altitude of 5 and 9 km with maximum difference of $\pm 0.3$ m s$^{-1}$. The CTL simulation overestimates the negative Doppler velocity by approximately 0.9 m s$^{-1}$ in the same range of altitudes compared to

the measured values. This result is consistent with the systematic underestimation of $N_i$ in CTR, and therefore, overestimation of the mean particle sizes and fall speeds.

## 6. Role of SIP in tropical convection

In the previous section it was shown that $w_{max}$ is higher in the SIP-4ICE simulation than in the control simulation above the melting layer (Fig. 6c), and that the inclusion of SIP results in enhanced formation of high IWC regions above the melting

layer. Altogether these results suggest that SIP plays an important role in the microphysics and thermodynamics of tropical convection, at least for the MCS examined. In order to explore SIP impacts in more detail, in this section, the rates and locations of SIP within the model storm and initiation of secondary convection are analyzed.

### 6.1. SIP productions at different altitudes

Figure 15 shows the active SIP areas and rates at different altitudes for the simulated MCS. Similar to Hu et al. (2021), the

values of $w$ are used to distinguish three different situations: 1) SIP within updrafts (marked by red lines, $w > 3$ m s$^{-1}$), 2) SIP within downdrafts (marked by yellow lines, $w < 3$ m s$^{-1}$), and 3) SIP with moderate vertical wind velocities (marked by blue lines, $-3$ m s$^{-1} \leq w \leq 3$ m s$^{-1}$).

For the employed SIP parameterizations, FFD is active in the range of altitudes $5 < H < 8.8$ km (Fig. 15d), whereas HM occurs in a narrower range of $5.2 < H < 6.8$ km (Fig. 15a). The range of the SIP activation altitudes are primarily determined

by the temperature ranges of the FFD and HM processes (section 2) and the temperature of the cloud base. The vertical velocity has a lesser effect on the SIP activation altitudes, and it may displace the upper and lower boundaries of the SIP regions within approximately $\pm 200$ m. Within these altitude ranges, SIP is mostly found in the area with moderate vertical wind velocities, followed by the area within updrafts, and occurs less in downdrafts for both HM and FFD processes (Fig. 15a, d). The active SIP areas of FFD are usually smaller than those of HM process in the range of altitudes $5.5 < H < 6.2$ km for all three situations.

To activate the FFD process, the employed parameterization requires the presence of large size raindrops (100 µm $< D <$ 3500 µm) together with the presence of small ice particles ($D < 100$ µm). This condition likely restricts the FFD process to a smaller area than that of HM process.





The vertical wind speed has a significant impact on the rate of SIP. Figure 15e shows the average SIP rates (m$^{-3}$ s$^{-1}$) within the active FFD areas shown in Fig. 15d. For most altitudes, the rate of FFD increases with an increase of $w$ and it is at least

one order of magnitude higher in updrafts than in the downdrafts. This is related to the lower number of precipitation-sized drops in downdrafts compared to updrafts.

Another factor is related to the effect of $w$ on the residence time of drops within the range of SIP activation altitude $\Delta H$. For a drop with a terminal fall velocity $V_{fall}(D)$ the residence time can be assessed as $\tau = \text{abs}\big(\Delta H/(V_{fall}(D) - w)\big)$. Depending on the sign of $(V_{fall}(D) - w)$ the drop will be moved upward through $\Delta H$ or downward. For the extreme situation

when $V_{fall}(D) = w$ the drop is suspended in an updraft indefinitely, it can freeze and generate secondary ice or mechanically interact with other cloud particles and thereby change its fall velocity.

The rate of the HM process (Fig. 15b) is higher in updrafts above 6.0 km where the riming process is active. At altitudes below 6.0 km, the rates are similar in updrafts, downdrafts and areas with moderate vertical velocities (-3 m s$^{-1}$ ≤ $w$ ≤ 3 m s$^{-1}$). Figure 15c shows the total SIP rate (m$^{-1}$ s$^{-1}$) from the HM process which is the product of the area of active HM and mean SIP

rate from HM shown in Figs. 15a,b. The total SIP rate shows how many ice particles are produced by SIP horizontally across the domain per 1 m of vertical layer per second. The total SIP rate from the FD process is shown in Fig. 15f. Below the altitude of 6.3 km, both the HM and FFD processes in the areas with moderate vertical wind velocities show the highest total SIP rate, followed by the area within updrafts. The lowest total rates are found in downdrafts. The larger active SIP areas associated with moderate vertical wind velocities (Figs. 15a, d) contribute significantly to the high total SIP rates across the domain. At

altitudes above 6.3 km, updraft regions contribute more to the total SIP rates. This is due to the high average rates in updrafts (Fig. 15b, e), since the corresponding SIP areas are smaller than those with moderate vertical velocities.

The results obtained show that the vertical extent $\Delta H$ of FFD is deeper and its rate is higher than those of HM. This finding leads to the conclusion that the overall contribution of FFD in the production of the secondary ice in tropical MCSs is significantly higher than HM.

**6.2. Role of SIP in initiating of secondary convection**

The freezing of rain into ice generates latent heating which should enhance the existing convection. This may explain the increase of $w_{max}$ above 6 km in the Figs. 6c and 7c in SIP-4ICE compared to CTR. As explained in the previous subsection, below 6.3 km altitude, both FFD and HM are more active outside of the major updrafts originating below the melting layer. High activity of the FFD and HM processes might eventually initiate new updrafts in stratiform regions inside MCSs above

the melting areas.

A Lagrangian trajectory analysis was used to trace the cloud parcels affected by SIP. For this analysis, 1152 air parcels were selected with active SIP at an altitude of 5.6 km from the SIP-4ICE simulation between 90 and 150 min. Each selected parcel is traced backward ($t < 0$ min) and forward ($t > 0$ min) for 15 min ($\Delta t = 30$ min). These parcels are then classified into two different groups. The first group included the parcels which at $t = -15$ min had altitudes within $5 < H < 6$ km and at $t = 15$



min ending their trajectories at $H > 6.5$ km. The second group included parcels with the same initial altitudes as the first group at $t = 0$ min. However, the altitude of their trajectories remained at $H < 6$ km at the end of forward tracing ($t = 15$ min). The total number of parcels of the first category was 47 and that of the second category was 1105.

Figure 16 shows the time history of mean values of environmental and microphysical parameters of the parcels for the two categories. Most parcels, which underwent SIP processes between -5 and 5 min, were located at the same altitude of 5.6 km at $t = 0$ min. These parcels started to rise from $t = -3$ min, and eventually reached 7.5 km at $t = 15$ min (Fig. 16c). On the other hand, the other group of parcels did not rise throughout the 30 min analysis period (hereafter named non-rising parcels). The mean altitude of these parcels decreased by about 700 m (Fig. 16c).

Figure 16e shows that the rising parcels had an initial positive vertical speed from $t = -3$ to -1 min. However, their potential temperature differences with respect to the environmental values at the same altitudes ($\Delta\theta$) were generally negative for $t < -1$ min, and decreased slightly from $t = -3$ to -1 min (Fig. 16g). Thus, these parcels were not gaining positive buoyancy at this stage. However, the $\Delta\theta$ of the rising parcels started to increase quickly between $t = 0$ min and 2 min, becoming positive and reaching a difference of nearly +1 K between $t = 3$ and 8 min (Fig. 16g). Figures 16i, j also show very high SIP rates of the rising parcels from the FFD process during this time period. For the non-rising parcels, there was an increase in $\Delta\theta$ during the same period but at a much slower rate due to smaller SIP tendency (Figs. 16i, and j). The $\Delta\theta$ remained negative during the whole analysis period, and these parcels were therefore convectively stable.

The main reason for the high SIP rate between $t = 0$ and 2 min for the rising parcels is that there was a substantial amount of large rain drops available for activating the FFD process (Figs. 16f, and h). The rain mean-mass diameter at $t = 0$ min was large (560 μm). The RWC was also large (0.36 g m$^{-3}$). The rain mean-mass diameter for the non-rising parcels (502 μm) was slightly smaller than that of rising parcels at $t = 0$ min. However, the corresponding RWC was quite low (0.02 g m$^{-3}$). With large RWC and rain mean-mass diameter, the rising parcels had a high SIP potential, which eventually led to increased greater freezing and increased latent heating and buoyancy, thereby enhancing secondary convection.

As mentioned in subsection 6.1, the FFD process plays a dominant role in SIP compared to the HM process. This agrees with what we found in Figs. 16i, j. The sudden increase of the $\Delta\theta$ with respect to the environment is more likely linked to the high SIP rate from FFD. SIP, in particular FFD, may therefore play a role in the initiation of new updrafts above the melting layer.

Figure 17 shows an example of a rising air parcel. The black line represents the parcel trajectory from $t = -15$ to 15 min. From $t = -15$ to 0 min, the air parcel had no significant change of altitude. Near $t = 0$ min, the air parcel was close to an existing updraft (Fig. 17a, red surface) and was in an area where there was rainwater (Fig. 17b, red surface). Shortly after $t = 0$ min, the air parcel started to rise. The supply of rainwater resulted in an enhanced SIP and led to a higher rate of latent heating and rapid increase the buoyancy.



## 7. Impact of ice aggregation

In addition to SIP, which clearly has significant effects on $N_i$, formation of high IWC, radar reflectivity, and vertical wind velocity, the aggregation of ice particles also plays an important role in determining the microstructure. Aggregation results in a decrease of $N_i$ and increase of radar reflectivity and particle fall velocity. The rate of aggregation is characterized by the ice-ice collection efficiency, $e_{ii}$. In the frame of this study, the following parameterization of the ice-ice collection efficiency has been employed (Cotton et al. 1986):

$$e_{ii} = \min(10^{0.035(T-237.16)-0.7}, 0.2) \tag{4}$$

where $T$ is the temperature in K.

There is a diversity of parameterizations of $e_{ii}$ employed in models (Khain and Pinsky, 2017), which may vary by up to three orders of magnitude. The uncertainty in $e_{ii}$, therefore, raises a question about the impact of the ice aggregation parameterization on the $N_i$ and high IWC formation.

To explore the effect of the ice aggregation rate on the high IWC formation, a sensitivity test (SIP-COL) was performed with another $e_{ii}$ parameterization, i.e.

$$e_{ii} = \begin{cases} 0.1; & \text{for T} < -20°C \\ 0.06T + 1.3; & \text{for} -20 < \text{T} < -5°C \\ 1; & \text{for} -5 < \text{T} < 0°C \end{cases} . \tag{5}$$

As seen from Eq. (5) the new $e_{ii}$ varied linearly from 0.1 to 1.0 within the temperature range $-20° < T < -5°C$. For $T < -20°C$, $e_{ii} = 0.1$, and for $-5°C < T < 0°C$ $e_{ii} = 1.0$. For all temperatures, $e_{ii}$ in (5) is much higher than that in (4).

Comparisons of distributions of $N_i$ for two SIP simulations with varying $e_{ii}$ and the observations are shown in Fig. 18. For the altitudes between 6 and 7 km, using the new parameterization of aggregation [given by Eq. (5)] (SIP-COL) results in a decrease of modal value of $F(N_i)$ by two orders of magnitude compared to using that described by Eq. (4) (SIP-4ICE). At higher altitudes between 10 and 11 km, the SIP-COL produces ~5 times higher $F(N_i)$ at $N_i$ of $10^5$ m$^{-3}$ and lower $F(N_i)$ by one to two orders of magnitude for $N_i > 4 \times 10^5$ m$^{-3}$ compared to those of SIP-4ICE.

Figure 19 shows similar results to Fig. 18 but for the distribution of ice mass. Between 6 and 7 km, applying the linear approach for $e_{ii}$ from Eq. (5) to the SIP-4ICE simulation (SIP-COL), the $F(IWC)$ between 0.4 and 2.4 g m$^{-3}$ is slightly reduced by up to 50%. However, the $F(IWC)$ for extreme situation with IWC larger than 2.5 g m$^{-3}$ is somehow slightly enhanced.

For altitudes between 11 and 12 km, the SIP-COL experiment produces a lower $F(IWC)$ up to ~1 order of magnitude for most of IWC ($> 0.16$ g m$^{-3}$) than the SIP-4ICE simulation. Although the estimation of $F(IWC)$ by SIP-COL between 6 and 7 km is relatively close to that of SIP-4ICE, SIP-COL produces much lower $F(IWC)$ in higher altitudes between 10 and 11 km than the SIP-4ICE simulation. This indicates that the $N_i$ at the lower altitudes play an important role in determining the IWC at upper altitudes.




## 8. Conclusion

The impacts of SIP on the microphysics and dynamics of deep convection have been examined using quasi-idealized near cloud-resolving simulations of a tropical MCS based on storm observations during the HAIC-HIWC field campaign. GEM model simulations using the P3 microphysics scheme were conducted using 250-m horizontal grid spacing and horizontally homogeneous atmospheric initial conditions, with updraft nudging to initiate convection. It was established through sensitivity tests that a minimum of three ice categories in P3 are necessary to examine SIP in detail; four categories were used for most of the simulations. P3 was modified to include rime splintering (HM) and fragmentation of freezing drops (FFD), which have been the most closely examined SIP mechanisms in laboratory studies. The parameterizations of the HM and FFD processes used were based on the information available from previously published results.

In the control configuration with no SIP processes at altitudes of 6 to 7 km, the simulated ice number concentrations were about two orders of magnitude lower than the values obtained from *in situ* measurements. The simulated frequency of encountering high IWC condition is about ½ to 1/500 of the observed frequency between IWC of 1 and 2 g m$^{-3}$. With the SIP mechanisms activated, the model results for these fields were dramatically improved compared to the observations. The Doppler velocities above the melting layer were also notably closer to the measured values, indicating improved ice fall speeds in the simulations with SIP active. SIP was responsible for an increase in ice concentrations of up to three orders of magnitude at altitudes of 6 to 7 km. As a result, the total ice mass was distributed over a much larger number of particles and thus mean particle size was smaller with a lower fall speed. Consequently, ice was more easily transported to higher altitudes, ultimately resulting in sustained cloud regions with high IWC.

Analysis of the simulations conducted lead to the following general conclusions:

1.      SIP processes play a critical role in the formation and maintenance of high IWC with low reflectivity at upper levels in MCSs.

2.      SIP enhances secondary convection above the melting layer due to an increase in buoyancy caused by greater latent heat releasing during vapor deposition on numerous secondary ice particles. Enhanced secondary convection may in turn extend the longevity of MCSs and regions with high IWC.

3.      Aggregation of ice particles results in a decrease of ice number concentration and IWC at upper levels but is very sensitive to details of the parameterization of this process, in particular the collection efficiency, which remains uncertain.

In order to minimize errors in interpretation of the results due to unresolved convective updrafts, the simulations conducted in this study were all done with a horizontal grid spacing of 250 m. This is a much higher resolution than current operational numerical weather prediction (NWP) models. However, tests with 1 km grid spacing (Fig. 20) indicated that impacts of including SIP are very similar to those at 250-m grid spacing, where 1 km is close to the grid spacing of several current operational and experimental NWP systems.  Further, the P3 microphysics scheme is already used operationally in the Canadian 2.5-km system (Milbrandt et al. 2015).  Therefore, the conclusions regarding the importance of including SIP





processes in models are not limited to numerical modelling in research mode, but also have important implications for current and/or upcoming operational NWP, in particular for systems that provide numerical guidance for civil aviation operating at
500    cruising altitudes between 10 and 14 km.

Finally, although the simulations conducted with the activated SIP process clearly resulted in improved results compared to the observations, this is not a basis for concluding that the HM and FFD parameterizations used are accurate representations of these physical processes. While the formulations were based on either laboratory experiments or combined modelling and *in situ* observational study, they are still largely *ad hoc*. This study further highlights the importance of these processes in deep
505    convection and the need to include them in some fashion in numerical models. However, accurate parameterizations that capture the underlying physics of these mechanisms, not just their bulk effects, continue to be topics of research.

**Code and Data Availability**

Observation data are available at https://data.eol.ucar.edu/master_lists/generated/haic-hiwc_2015 (last access: 31 March 2022).

510    **Authors Contribution**

ZQ, AK and JAM conceptualized the research goals and aims. ZQ, JAM and AK designed the experiments with the support from YH, GMM and HM. ZQ performed the simulations and analysis with the help from AK, JAM, IH, MW and CN. ZQ prepared the manuscript with contributions from all co-authors.

**Competing interests**

515    The authors declare that they have no conflict of interest.

**Acknowledgement**

The authors thank Manon Faucher and Melissa Cholette for their help with setting up and running the GEM model. The authors also thanks Wei Wu for his help with reviewing the manuscript and Alfons Schwarzenboeck for providing data from Falon-20 aircraft.

520    **Financial support**

The HAIC-HIWC program was supported by the Federal Aviation Administration (FAA), European Aviation Safety Administration (EASA), Environment and Climate Change Canada (ECCC), National Research Council (NRC), and Transport Canada (TC). GM was supported by the National Science Foundation (grant no. 1842094).



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

**Table 1: Summary of GEM configurations details. References to specific schemes are provided in Milbrandt et al. (2015).**

| Dynamics/numerics |
|---|
| • Nonhydrostatic primitive equations |
| • Limited-area grid on a latitude–longitude projection |
| • Uniform horizontal grid spacing of 0.00225 longitude (approx 0.25 km) |
| • 82 vertical levels |
| • Upper-boundary nesting above 10 hPa |
| • Time step of 15 s |
| • Terrain-following Gal-Chen vertical coordinate |
| • Two-time-level semi-implicit time differencing |
| • 3D semi-Lagrangian advection |
| • $\nabla^4$ horizontal diffusion ($\nabla^6$ for potential temperature) |
| **Physics** |
| • Planetary boundary layer scheme based on turbulence kinetic energy with statistical representation of subgrid-scale cloudiness (MoisTKE) |
| • Kuo–transient shallow convection scheme |
| • P3 two-moment bulk microphysics scheme |
| • Li–Barker correlated-k distribution radiative transfer scheme (called every 3 min) |
| • Interaction Sol-Biosphère-Atmosphère (ISBA) land surface scheme |
| • Distinct roughness lengths for momentum and heat/humidity |



**Table 2: List of simulations.**

| Experiment name | Coalescence efficiency between ice particles | Number of ice category | Secondary Ice production |
|---|---|---|---|
| BASE-1ICE | Cotton et al., 1986 | 1 | No SIP |
| BASE-2ICE | | 2 | |
| BASE-3ICE | | 3 | |
| BASE-4ICE (CTR) | | 4 | |
| BASE-COL | Linear from 0.1 to 1.0 between -20°C and -5°C. Below -20°C: 0.1, above -5°C: 1.0 | 4 | |
| SIP-2ICE | Cotton et al., 1986 | 2 | Droplet shattering: large rain droplet (100 µm < $D$ < 3500 µm) collected by small ice particle ($D$ < 100 µm) |
| SIP-3ICE | | 3 | |
| SIP-4ICE | | 4 | |
| SIP-COL | Linear from 0.1 to 1.0 between -20°C and -5°C. Below -20°C: 0.1, above -5°C: 1.0 | 4 | Hallett-Mossop: applied to the remaining collected rain by ice |


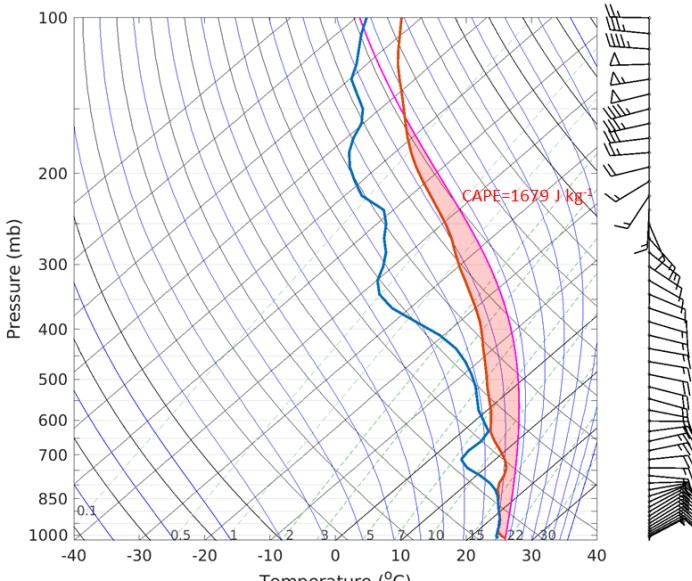

**Figure 1: Initial atmospheric profiles for the idealized simulations. Blue line: dew point temperature. Red line: environment sounding (temperature). Magenta line: parcel lapse rate.**



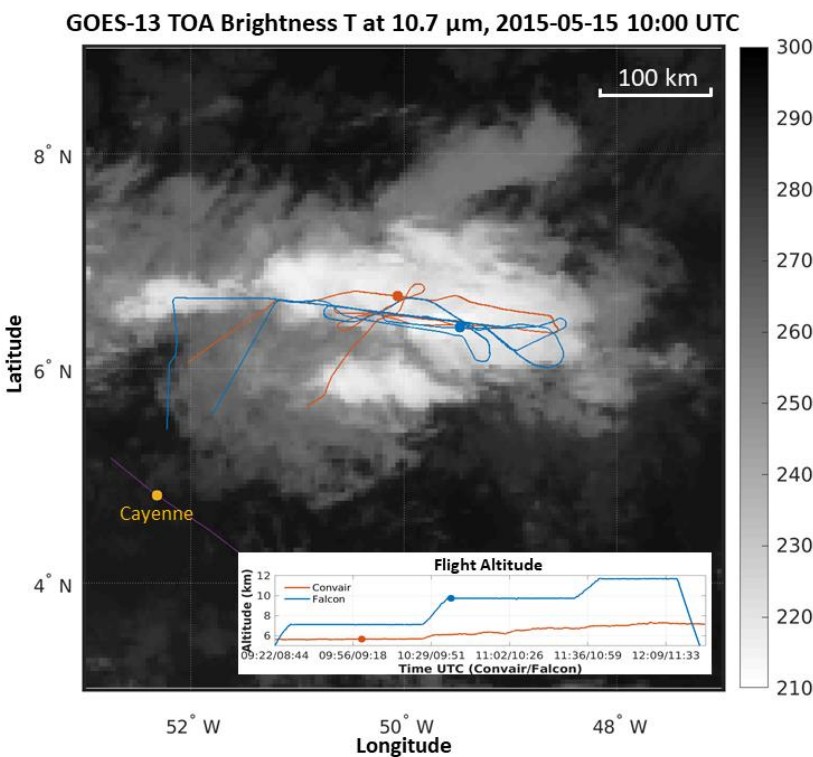

**Figure 2: GOES-13 top of atmosphere (TOA) brightness temperature at 10.7 $\mu$m at 1000 UTC 15 May 2015 (Knapp et al. 2018). Blue lines: Falcon-20 flight track and altitudes. Red lines: Convair-580 flight track and altitudes. Red and blue dots indicate the locations and altitudes of the Convair-580 and Falcon-20 aircraft, respectively, at 1000 UTC. Purple line: coastline.**

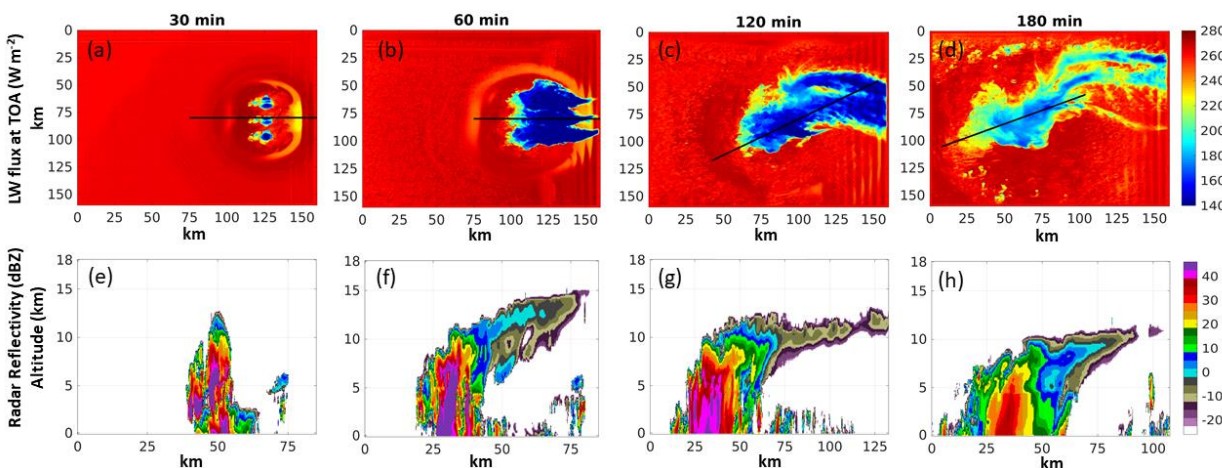

**Figure 3: Simulation with a single ice category in the P3 bulk microphysics scheme. a-d: upward longwave radiative flux at the top of atmosphere at for different times (30, 60, 120, 180 min after the model initiation). e-h: the corresponding radar reflectivity of the cross-section marked by the black line in a-d.**



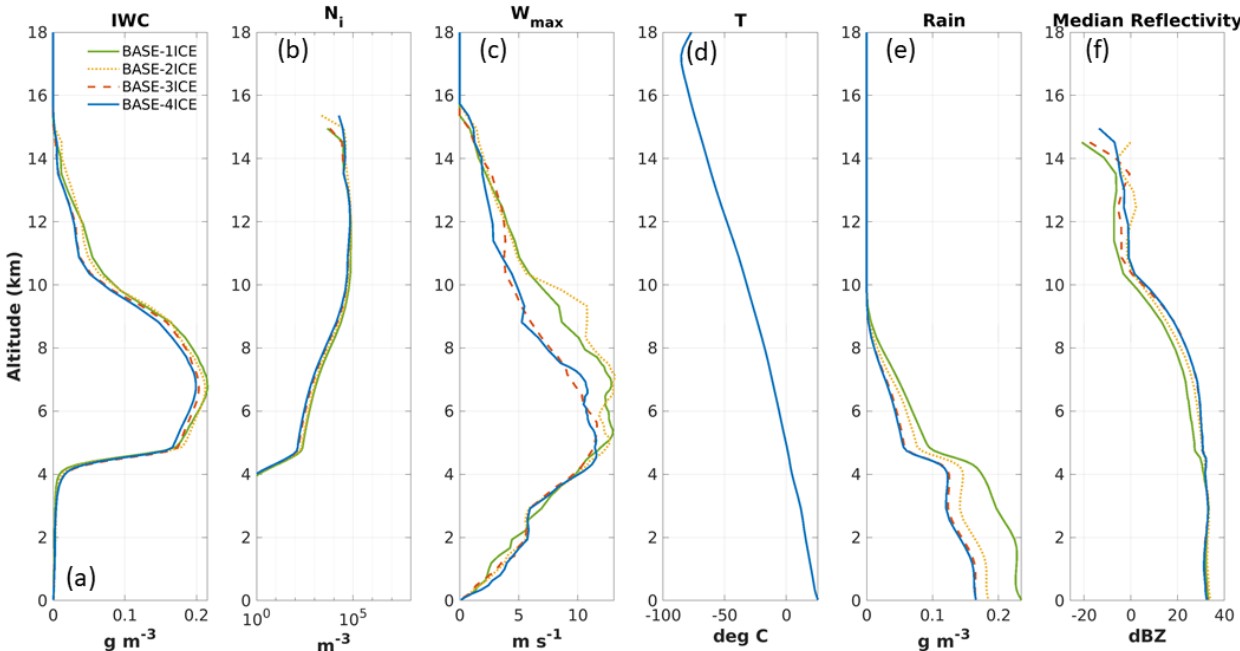


**Figure 4: Profiles from the baseline simulations without SIP. (a): ice water content (IWC), (b): ice number concentration $N_i$, (c): maximum vertical wind speed $W_{max}$, (d) air temperature (the four simulations have only slightly different temperature that is not distinguishable in the figure) T, (e): rain water content RWC, (f): median radar reflectivity. The profiles are calculated based on data from 90 to 150 min of simulation. (a) and (b) are horizontally averaged over regions with IWC > 0.001 g m$^{-3}$, (c) is the horizontal**

**maximum across the domain, (d) is a horizontal average over the whole domain, (e) is an average over the area with both ice water path and rain water path larger than 1 g m$^{-2}$, and (f) shows median values including points with either IWC or RWC larger than 0.01 g m$^{-3}$ within the mask used for (e).**





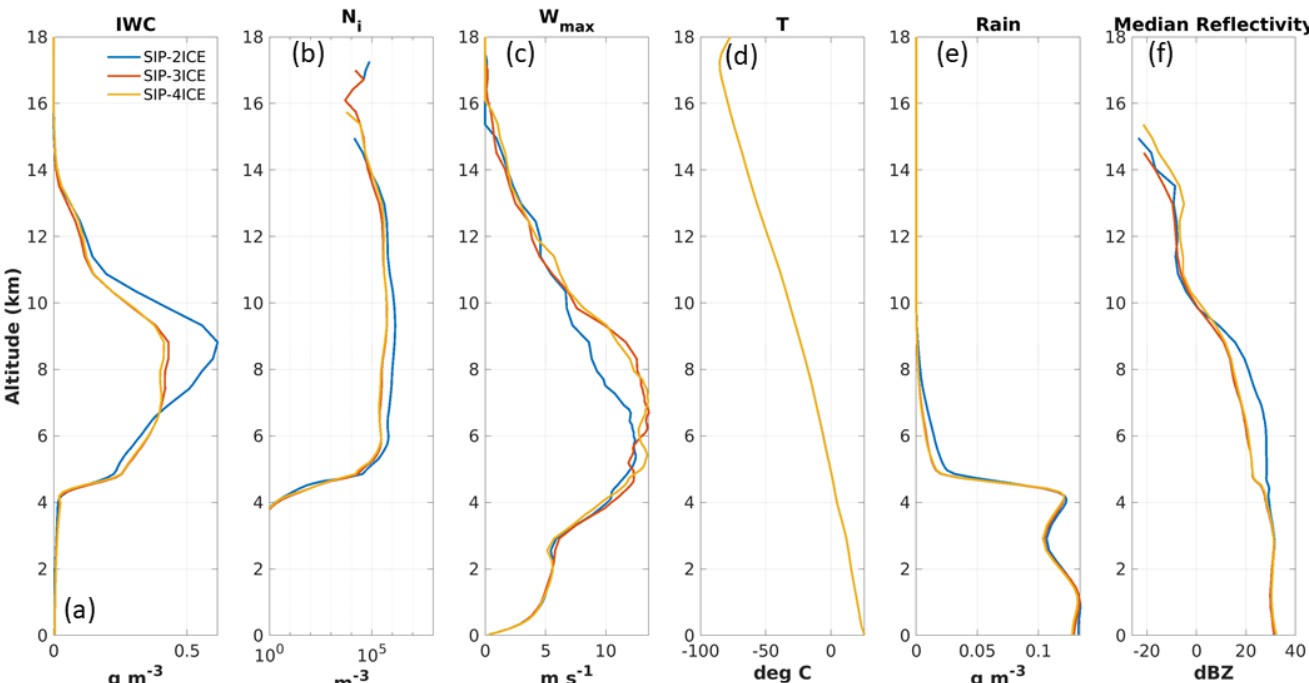

**Figure 5: same as figure 4 but for SIP simulation with 2, 3 and 4 ice categories.**

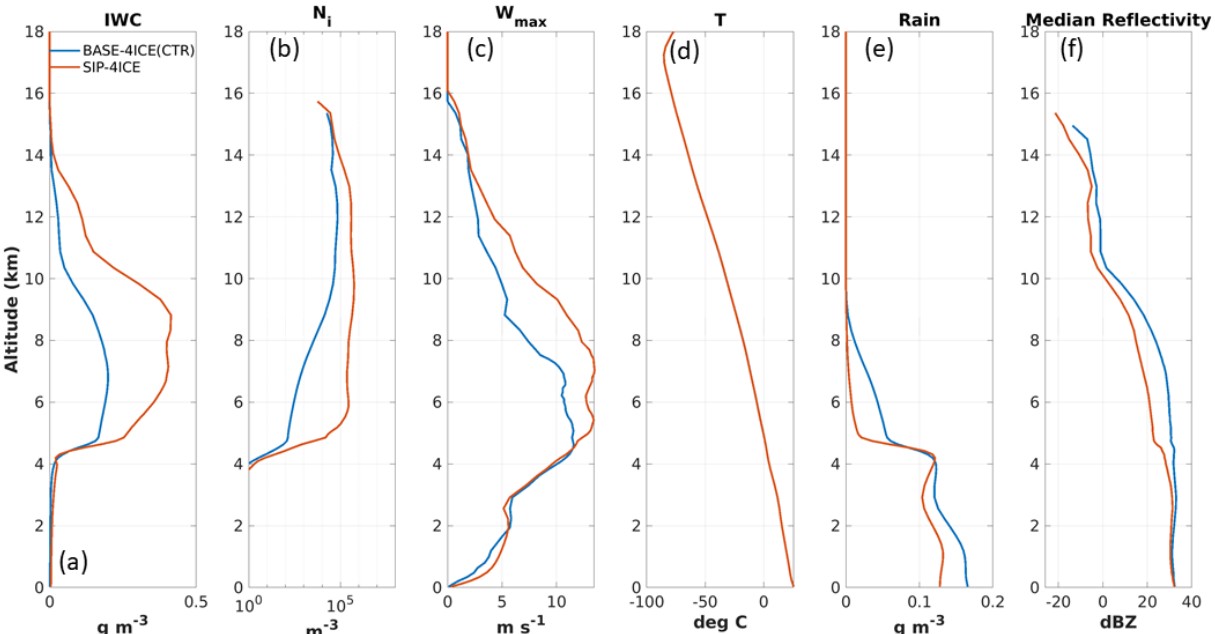

**Figure 6: Same as Figure 4, but for baseline (CTR) simulation with 4 ice categories and the SIP simulation with 4 ice categories.**



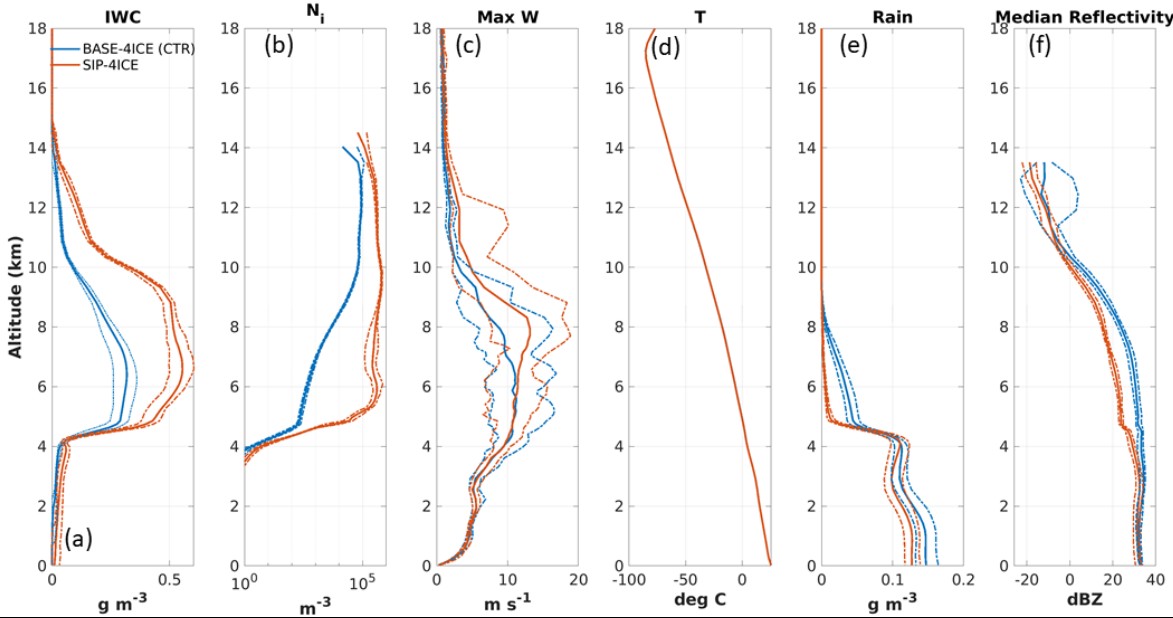

**Figure 7: Same as Figure 4. Domain averaged profiles for the CTR and SIP simulations with 4 ice categories, for 11 ensemble members each. The dash-dot lines show the minimum and maximum values among the ensemble members. The results are calculated based on data at 120 min after the model initiation.**



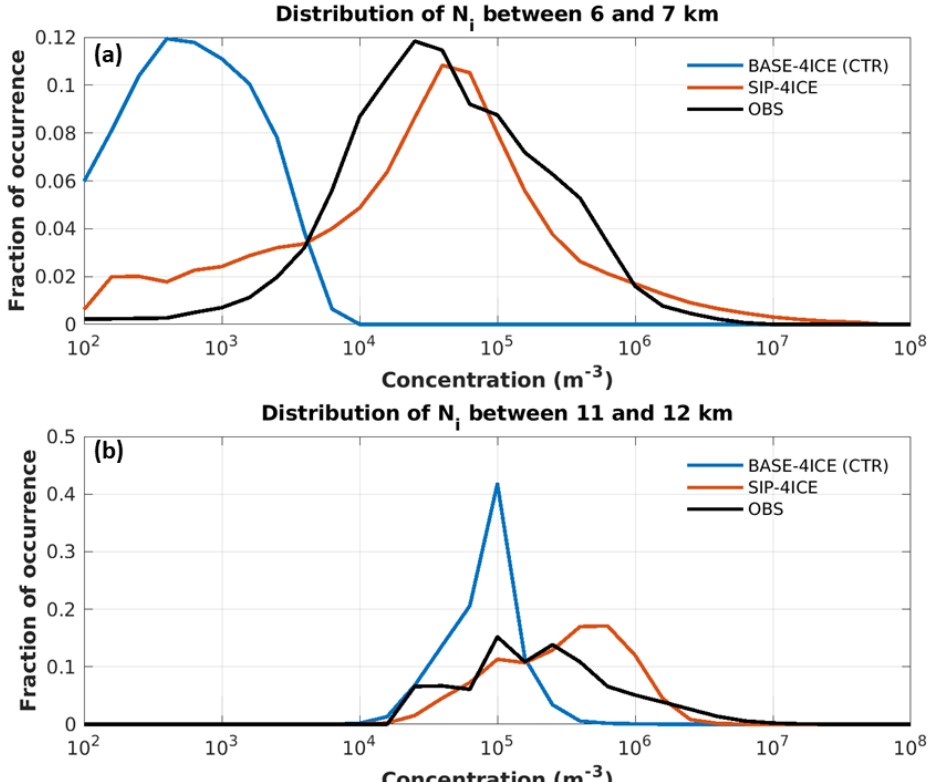

**Figure 8. Distribution of $N_i$ for model simulations and for the observation data from the HAIC-HIWC aircraft campaign near French Guiana in May 2015. Logarithmic bin width of 1/5 of an order of magnitude is used. (a): results from data between altitudes of 6 and 7 km and with ice water content higher than 0.01 g m$^{-3}$. (b): same as (a) but for altitudes between 11 and 12 km.**



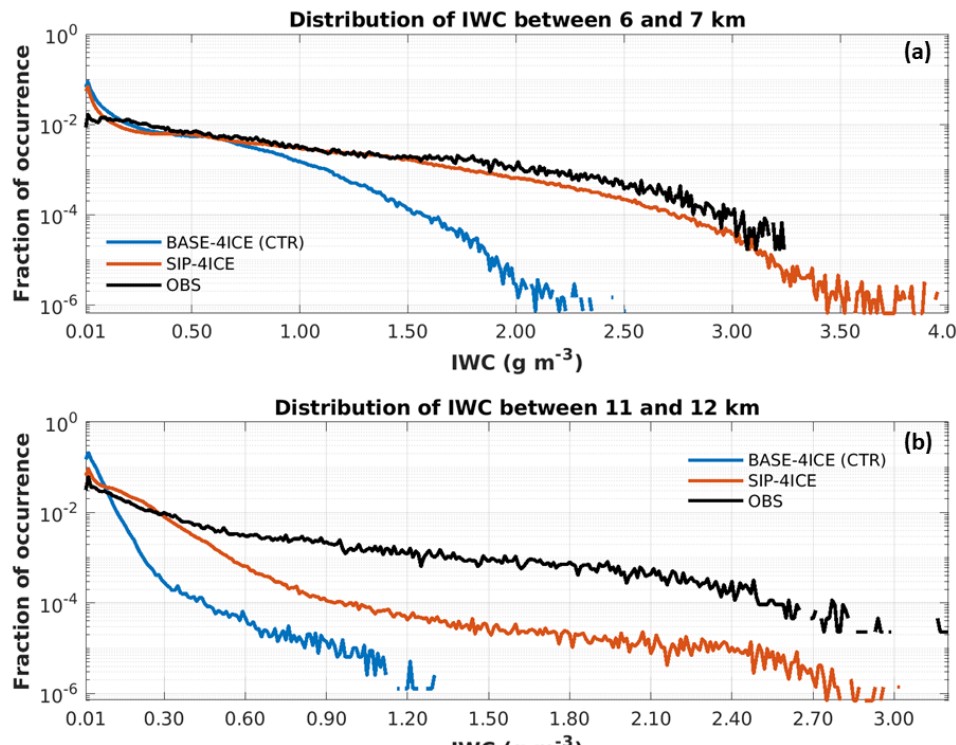


**Figure 9. Distributions of ice water content from the model simulations and observation data from the HAIC-HIWC aircraft campaign near French Guiana in May 2015. The bin width of 0.01 g m⁻³ is used. (a): between the altitude of 6 and 7 km with observation data from the NRC Convair-580 aircraft, (b): between 11 and 12 km with the observation data from the SAFIRE Falcon-20 aircraft.**






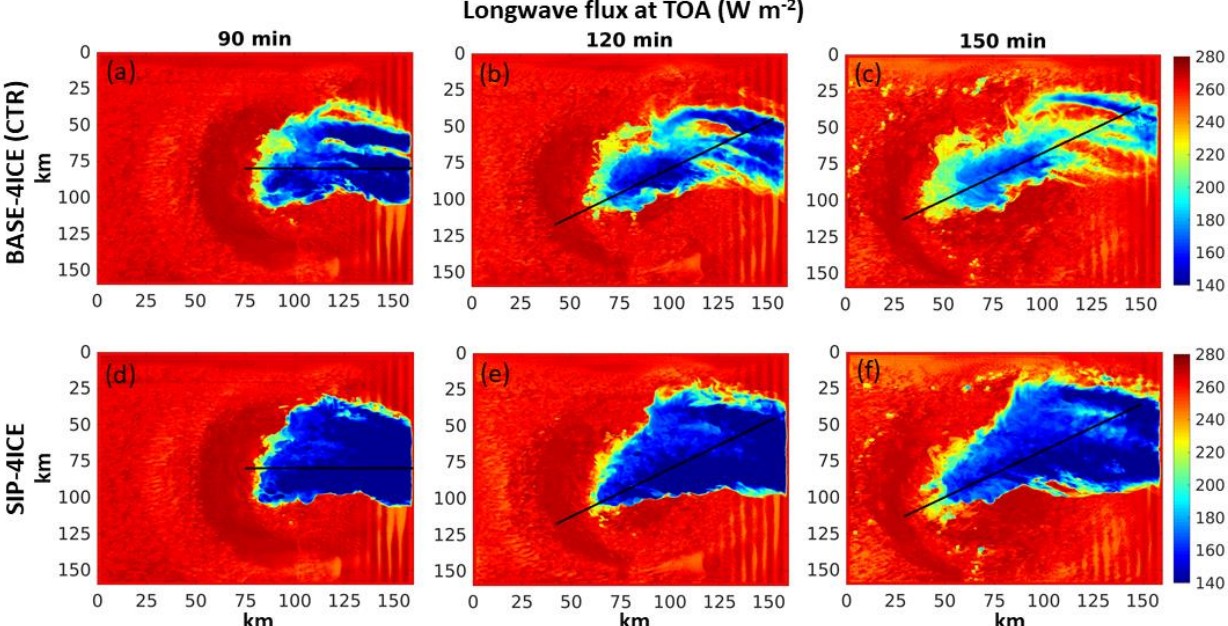

**Figure 10: Longwave flux at the top of atmosphere. a-c: simulation with baseline GEM set-up with 4 free-categories of ice at 60, 120 and 180 min after the initiation. d-f: same as a-c but for the simulation with secondary ice production implemented. Black lines indicate the location of the cross-sections shown in later results.**


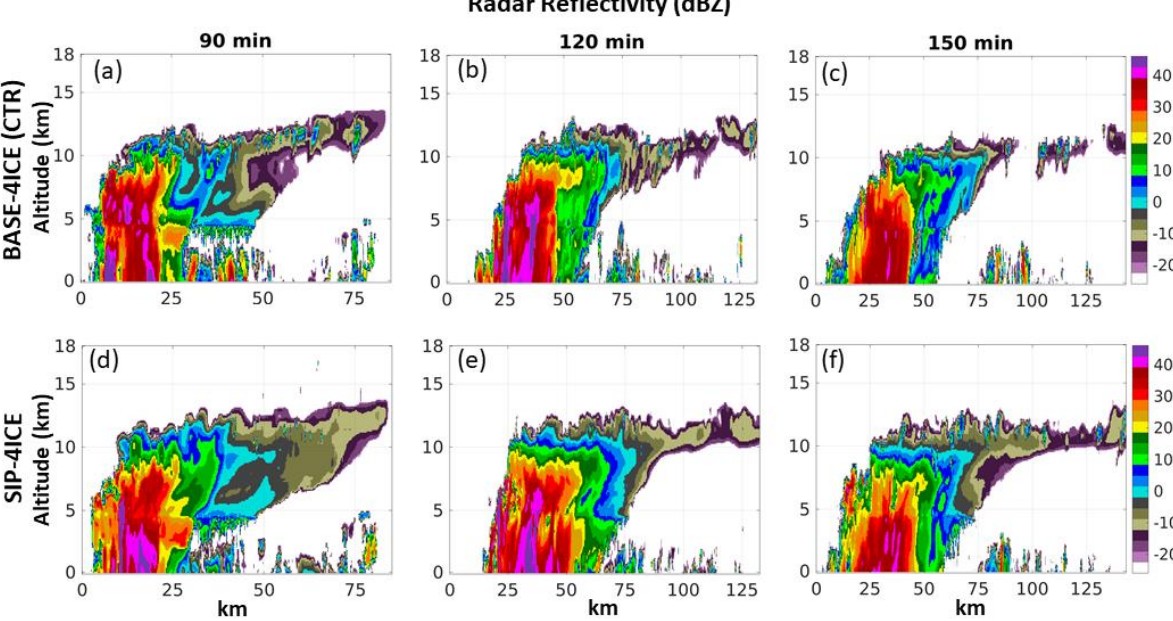

**Figure 11: Simulated radar reflectivity for the cross-sections indicated by the black lines in Figure 10.**





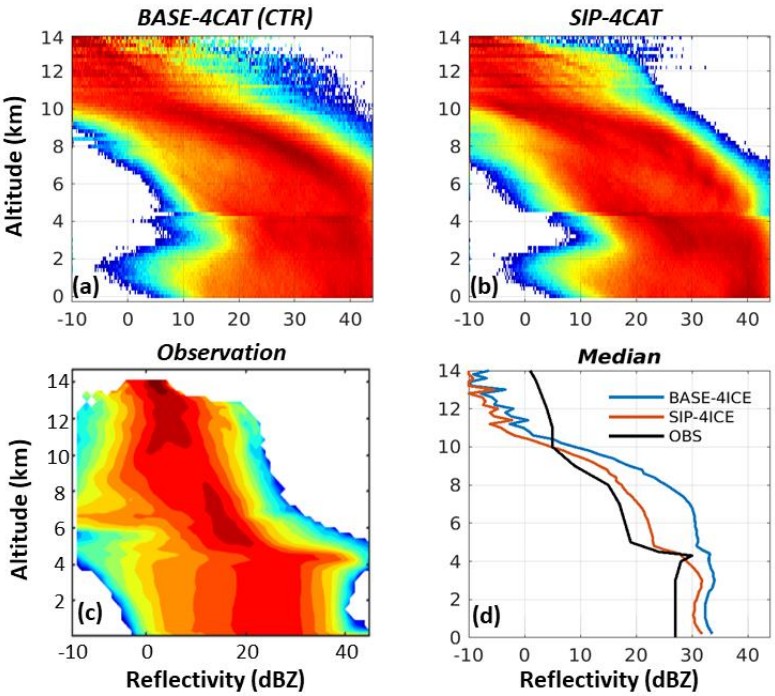

**Figure 12: Radar reflectivity distribution frequency for (a): CTR with 4 ice categories between 60 and 180 min after model initiation, (b): same as for (a) but for the model with SIP included, (c): from all the observation data from the NRC Convair-580 aircraft during the HIAC-HIWC campaign, and (d): the median value for each altitude for the two model simulations and observations.**

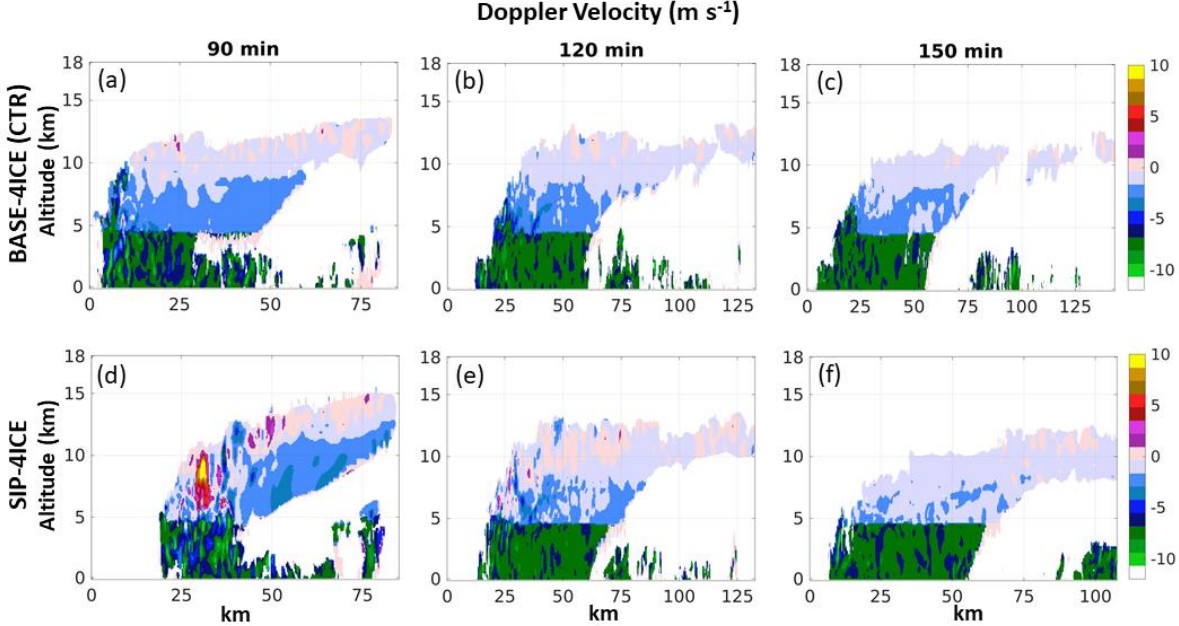

**Figure 13. Simulated Doppler speed for the cross-sections indicated by the black lines in Figure 10.**



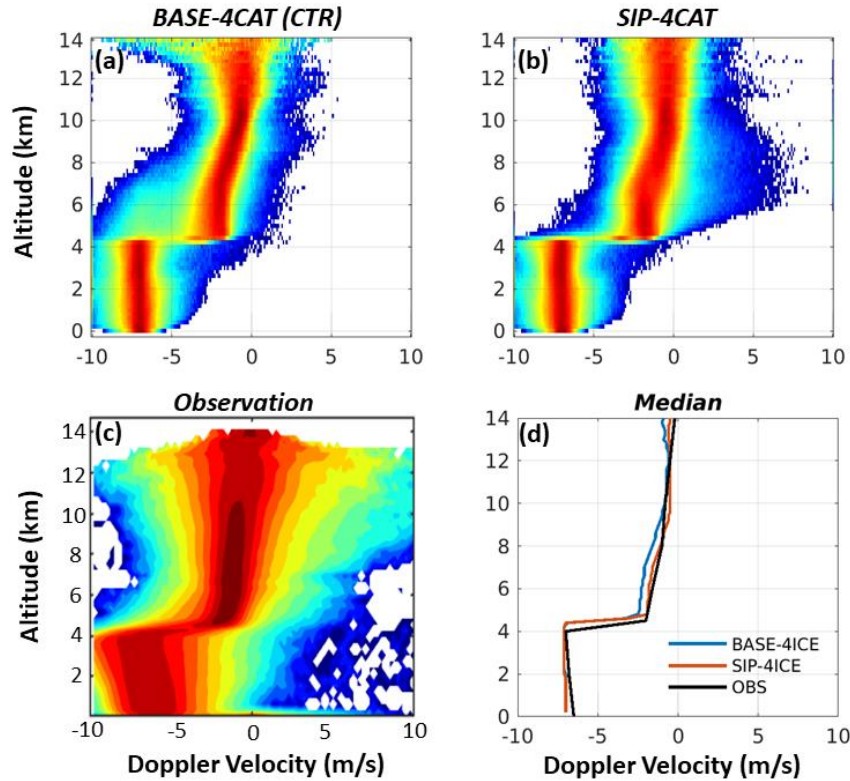

**Figure 14. Same for Figure 12 but for the Doppler velocity.**






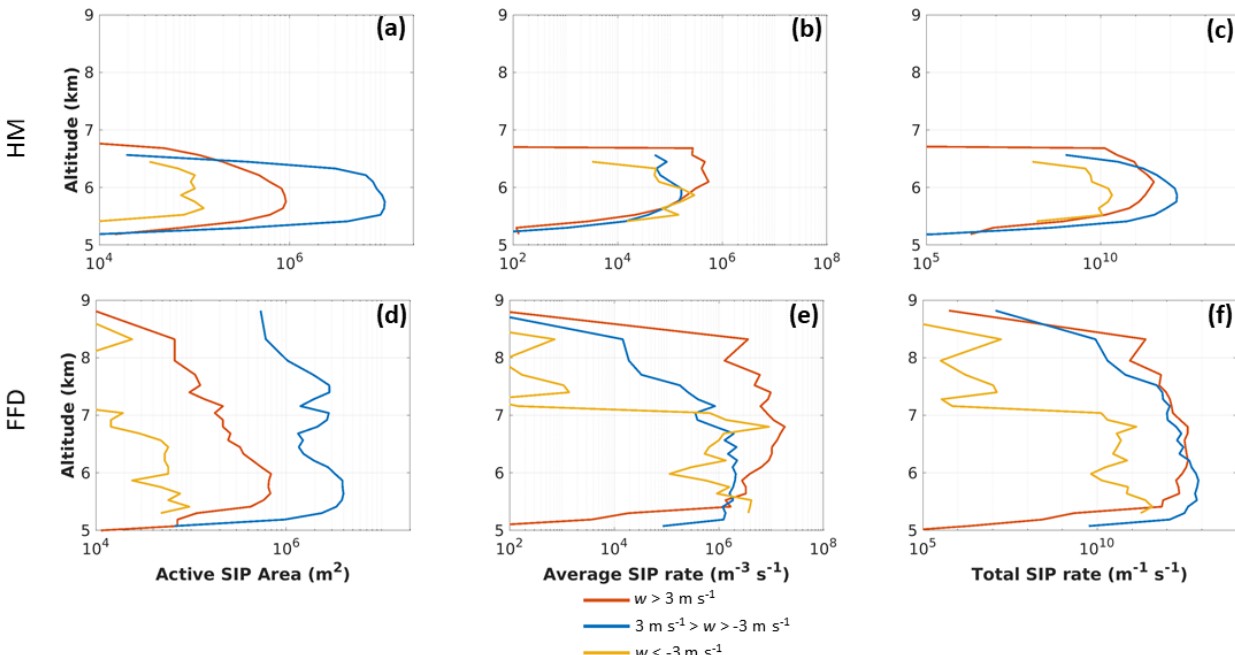

**Figure 15. (a): area in m² with active HM process; (b): average rate of SIP by the HM process within the active area shown in (a); (c): total SIP rate, product of (a) and (b). (d) to (f): same as (a) to (c) but for the FFD process. Red lines (updrafts): SIP with $w > 3$ m s⁻¹, blue lines (outside of updrafts/downdrafts): SIP with $w$ between -3 and 3 m s⁻¹, yellow lines (downdrafts): SIP with $w < -3$ m s⁻¹. All results are temporal averages between 90 and 120 min.**




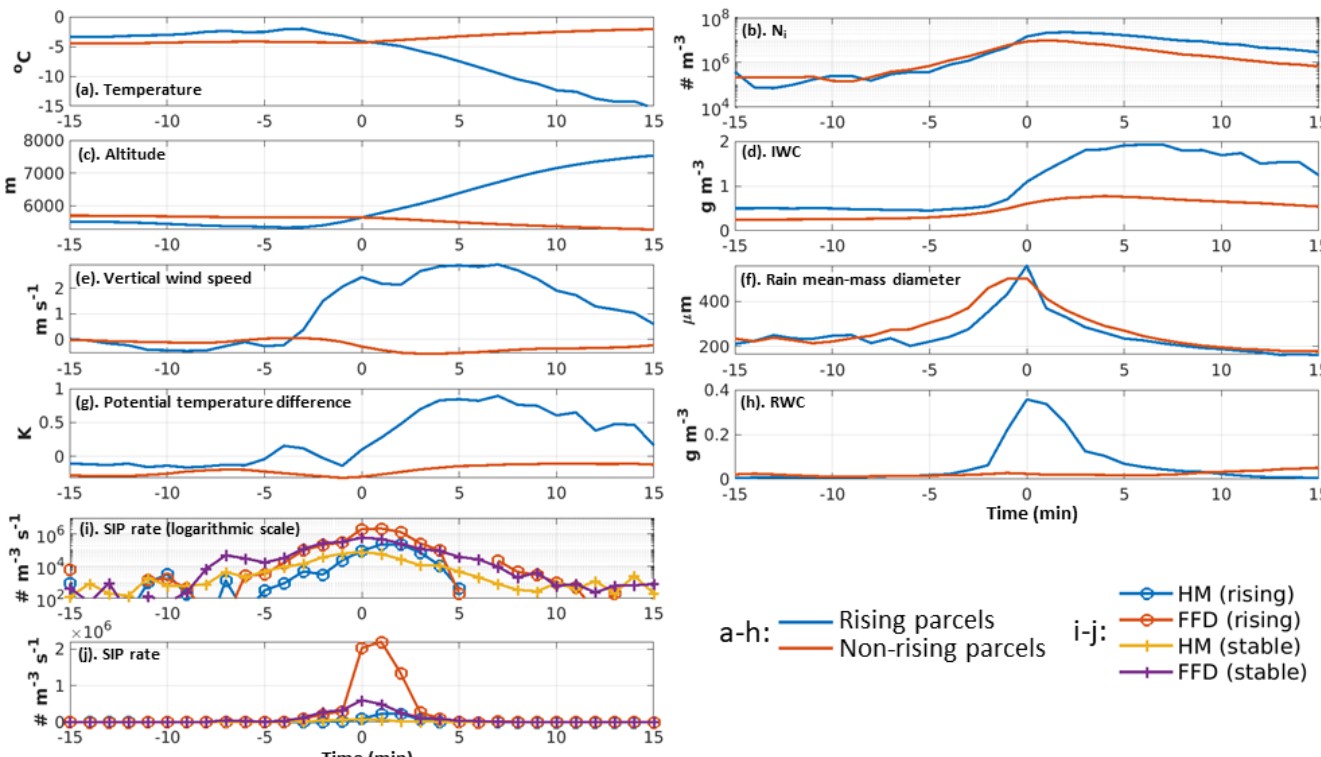

**Figure 16: trajectory tracing of different variables for the rising parcels (blue lines in a-h) and the non-rising parcels (red lines in a-h). a: temperature, b: Ni, c: Altitude, d: IWC, e: vertical wind speed, f: rain mean-mass diameter, g: Δθ, h: RWC, i: SIP tendency in logarithmic scale, j: SIP tendency in regular scale.**




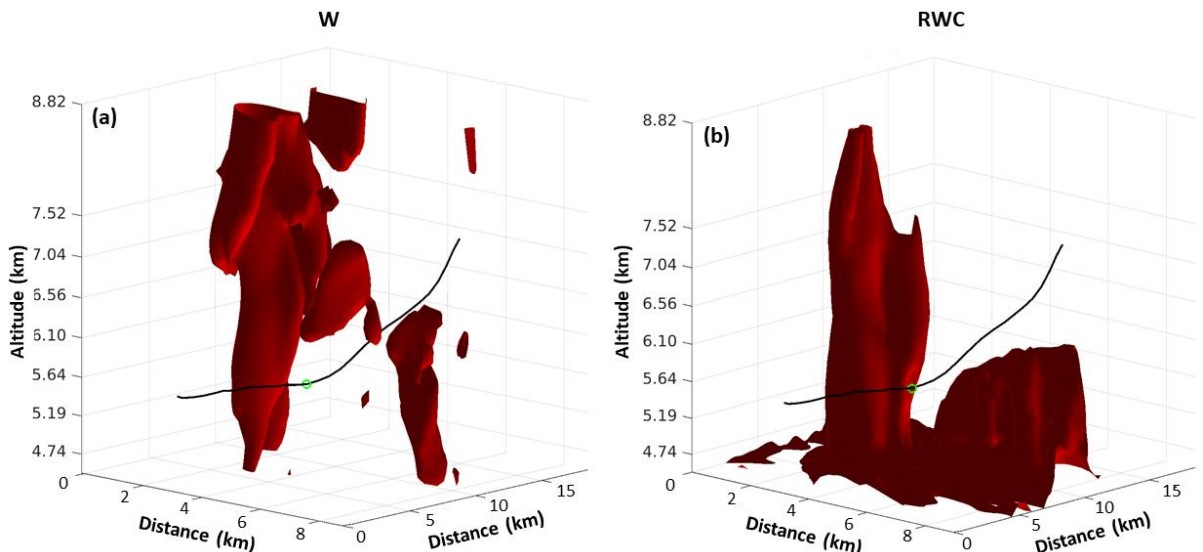

**Figure 17: the trajectory of a specific air parcel (black line) and the location of the parcel at $t$ = 0 min (green circle). The red surfaces in (a) show the area with $w$ larger than 3 m/s, and in (b) with RWC larger than 0.05 g m$^{-3}$, at the moment of $t$ = 0 min.**

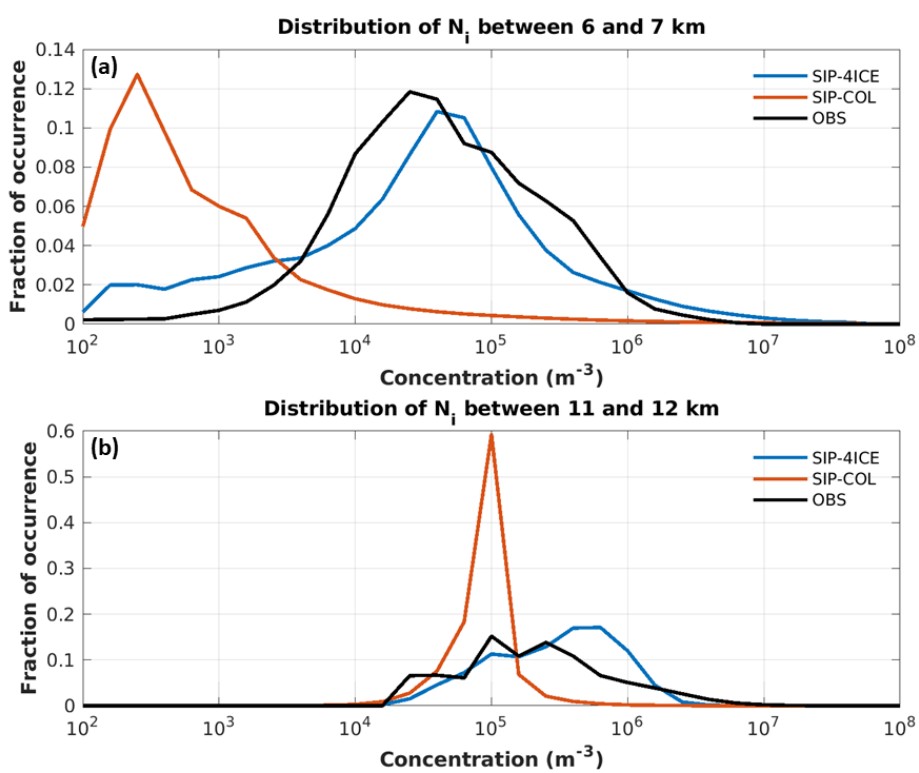


**Figure 18: Similar to Figure 8, but for different simulations: SIP-4ICE and SIP-COL.**





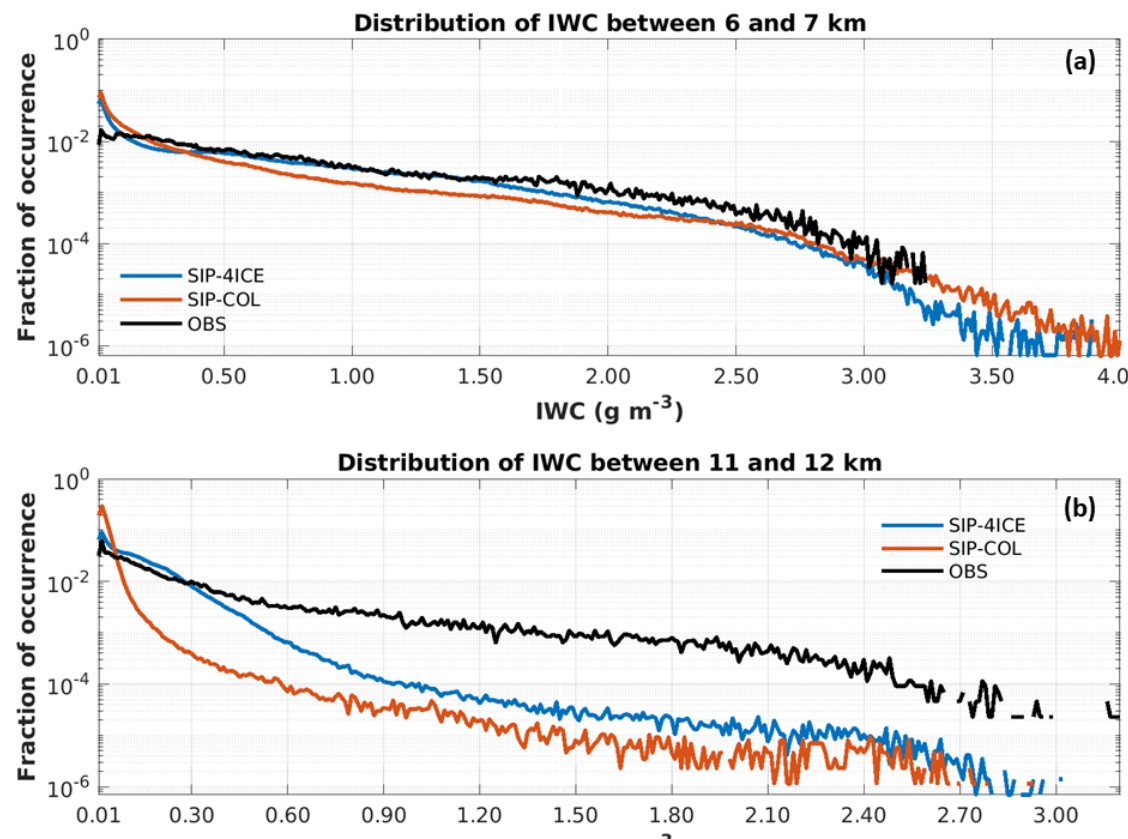

**Figure 19: Similar to Figure 9 but for different simulations: SIP-4ICE and SIP-COL.**




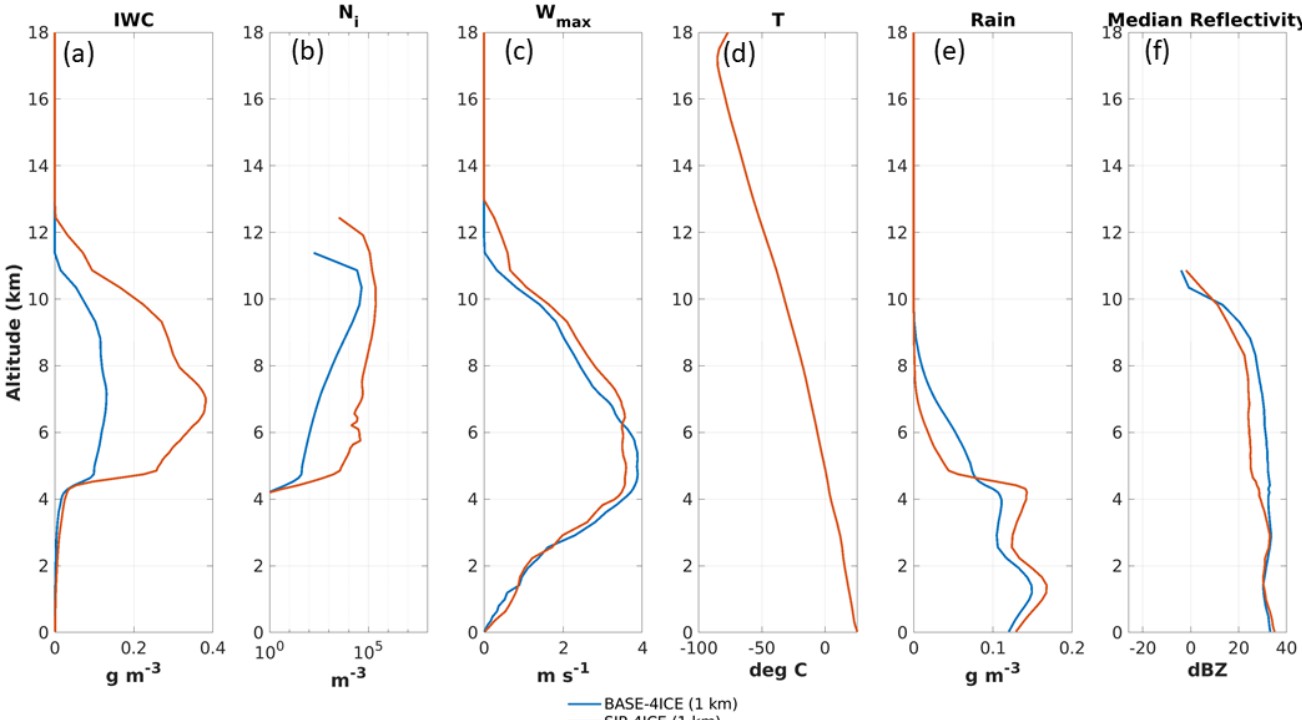

**Figure 20: Similar to Figure 6 but for simulation at 1 km horizontal grid spacing.**