# Peer review of "The impacts of secondary ice production on microphysics and dynamics in tropical convection"

_EGUsphere, 2022_

## Referee Comment (RC2)

Review of "The impacts of secondary ice production on microphysics and dynamics in tropical convection", by Qu et al.

This paper investigates the importance of two SIP processes on tropical convection using high-resolution simulations. They also make use of observations from the HAI-HIWC program to provide some evaluation of the performance of the simulations.
I really enjoyed reading this paper, which is very well written and organized. I have a few comments, mostly regarding the comparisons with the observations, which would need to be addressed before the paper is ready for publication. These comments are listed below.

Main comments

- In the paragraph starting line 299, the authors comment that the differences in IWC statistics at 11-12 km could be due to differences in the sampling of data, with the HAIC-HIWC program targeting conditions with high IWC. However, the same comment applies to all comparisons in this paper, including those that seem to work like IWC at 6-7 km, and number concentrations. I think it is the main weakness of the paper not to have attempted to minimize these differences in sampling. There are different ways to address this problem. What I suggest would make more sense is to compare relationships between parameters. For instance, Ni versus reflectivity, IWC versus reflectivity, at different heights (6-7 and 11-12km like you've done is good) so that you can compare in different ranges of reflectivity how the Ni and IWC distributions compare with the observed ones. Or use vertical velocity like you've done later in the paper when looking at SIP rates. I am very confident that with a little more care you will be able to say more about how well the simulation captures what the observations say.
- Results from Fig. 6, especially IWC and Z: This is a good illustration of my previous point, Differences in IWC and Z should result in very different IWC-Z relationships from the two simulations. You could compare simulations in IWC – Z space and compare with observations of IWC and Z (simulated from PSDs or measured by NAX close to the aircraft) from HAIC-HIWC too. Regions of simulated storms that were not measured by the aircraft will not be covered by observations, but that's OK, just compare where you can in the range of possible IWC and Z values.
- Fig. 12: The Convair flew slightly above or slightly below the 0degC isotherm. If not corrected for attenuation, your stats from the Convair cannot be used as a reference. Has attenuation been corrected? I suspect not. A maximum of radar Z in liquid of ~25 dBZ with a maximum at ~ 40 dBZ at the right side of the CFAD seems too low for tropical convection at X-band.
- Comparisons from Fig. 12: I think there is more to say about this figure. I also think the colour scale of your reflectivity CFADs is not well chosen it does not highlight enough the maxima that are of most interest:
  - just below the melting layer there is a local maximum of rainfall in the SIP simulation at ~ 25 dBZ, where the observations show a maximum. It corresponds to a blob of higher frequencies ion the CFAD produced by SIP around +15 dBZ above the melting layer. It is very impressive! Probably the most important result I see in those comparisons.
  - The SIP simulation still produces very large ice particles (30 dBZ+) in the HM region that are completely absent from the observations. That is a long-standing problem in tropical convection simulations, it needs to be mentioned and discussed. Used to be associated with way too much graupel produced by the models. But in your case you don't use the same type of parameterizations, so what causes this population of ice particles to be there ? Calls from a deeper analysis of your four ice categories ? Is there one in particular that

occupies this space on the reflectivity CFAD ? How are the four ice categories spread out on the CFAD ? That would be ar eally interesting plot to make and analyze, maybe.

- Following up from the previous comment, nothing is done to show how the four ice categories end up looking like and how they contribute to the total Ni, IWC, etc ...

Other comments:
Line 168: " There have BEEN" ("been" is missing I think)

Line 177: Mossop (not Mossp)

Section 4, lines 239 - 240: it would be interesting to document somehow what difference this choice makes to the morphology of the storm for SIP simulations. Would it be possible to compare cross sections (horizontal or vertical) as well ? Or comment on how much that choice changes the morphology of the storm without showing figures? Seems like a good opportunity.

Figure 7: The only parameter that shows much less difference between simulations in the ensemble exercise is vertical velocity. Any idea why ? You should offer more explanations about that, and acknowledge that as well.

Lines 285-288: Here you need to talk about the width of the distribution. Although the max in obs is incidentally close to the CTR simulation, the width of the distribution is much better captured by the SIP simulation. Then you need to investigate why there is this high population of $10^6$ Ni in the simulation, maybe with some sensitivity tests ? If it's good at 6-7 km, but not at 11-12 (higher concentrations), it seems unrelated to how well you simulate SIP maybe?

Section 5.4 : differences in LW radiation are really interesting and show substantial differences between non-SIP and SIP comparisons. Why didn't you attempt to compare with satellite products?

Lines 367-368: If I understand correctly panel a is not normalized by the actual areas of the three updraft categories. Maybe the lower values for downdrafts is just because there is much less area actually occupied by downdrafts? I wonder if it would be complementary to look at an activation fractional area (% of downdraft area where SIP is doing something)?

Lines 392-395 and line 427: This reads like you conclude it's like that in real life tropical convection. Most of this is driven by the choices you made for the parameterization, so maybe this comment should be restricted to "in these simulations ...". The same caution should be exercised in the general conclusions, I think.

Line 451: We sort of discover here that your objective is to test whether a higher collection efficiency has large influence on the result. I think you should state clearly in lines 444-448 that you are taking an extreme case of higher efficiency to explore. This also calls for another question: why did you chose the default one you used? Is it known to be more physically realistic?

Fig. 20: it is striking that the vertical velocities are a lot lower at 1km compared to 250m. This should be mentioned I guess? Also I think it would be interesting to add the profiles of Fig. 6 (as dashed line) on Fig. 20 to support this discussion.

Good luck with the Review
Alain Protat
Melbourne, 20/06/2022

---

## Author Response (AR1)

**Answers to the reviews of "the impact of secondary ice production on microphysics and dynamics in tropical convection" by Qu et al.**

**Reviewer 1:**

The topic of this paper is very timely, with a recognised need to study and quantify the impacts of SIP. Tropical convection in particular is important to quantify as the tropics are thought to be a region where cloud-climate feedbacks may be substantial, and there are many uncertainties. The approach is also state-of-the-art with high-resolution modelling and observations. I think the paper is a worthy topic for ACP, and has some important findings so should be accepted, but I felt a few points needed to be addressed. An important part is bringing the important evidence to the forefront of the paper so that it is very clear to the reader. I have made some suggestions below that may help.

Thank you very much for your positive review! We answered your questions in paragraphs written in blue color.

**General**

The paper presents compelling evidence that SIP mechanisms are important in a tropical MCS. If I have understood correctly I think the strongest evidence is presented in Figures 8 (histograms of Nice); perhaps Figure 9a; and Figure 18 and 19a.

If this is the case, and you agree with me, could the paper be arranged to highlight these points so that it is clear what the main evidence is? E.g. the large ice crystal number mode seen in histograms of the observational data could only be reproduced with SIP switched on in the model. And the broad distribution of IWCs, extending to large values could only be reproduced with SIP switched on. Maybe this could be highlighted in the abstract?

We made the following change in the abstract to highlight the points (in bold):

*"Secondary ice production (SIP) is an important physical phenomenon that results in an increase of ice particle concentration and can therefore have a significant impact on the evolution of clouds. In this study, idealized simulations of a mesoscale convective systems (MCS) was conducted using a high-resolution (250-m horizontal grid spacing) mesoscale model and a detailed bulk microphysics scheme in order to examine the impacts of SIP on the microphysics and dynamics of a simulated tropical MCS. The simulations were compared to airborne in situ and remote sensing observations collected during the High Altitude Ice Crystals - High Ice Water Content (HAIC-HIWC) field campaign in 2015. **It was found that the observed high ice number concentration can only be simulated by the models which include SIP processes. Inclusion of SIP processes in the microphysics scheme is crucial for the production and maintenance of high ice water content observed in tropical convection.** It was shown that SIP can enhance the strength*

*of the existing convective updrafts and result in the initiation of new updrafts above the melting layer. Agreement between the simulations and observations highlights the impacts of SIP on the maintenance of tropical MCSs in nature and the importance of including SIP parameterizations in models."*

There is a question though because the model fails to produce larger concentrations and water contents at the higher altitudes (11-12 km). Hence, the question I would ask is, is the modelled storm strong enough compared to the observation. If the model were more intense you may find that more ice would be present in both the 6-7 km bin and the 11-12 km, so you may not require quite as much SIP to explain the observations. I think it is worth considering.

Very Good question. Apart from this study with idealized simulations, we also conducted some real case simulations with both BASE-3ICE and SIP-3ICE configuration over Cape Verde (CADDIWA field campaign, https://www.safire.fr/en/content_page/campagnes-en-cours/caddiwa.html). From these real case simulations, we had very similar conclusions: baseline simulations significantly underestimated the IWC and $N_i$, whereas SIP simulations gave much better agreements to the observation data. This might be an indirect confirmation that the idealized simulations used in this study dose give, to a certain degree, reasonable results for the tropical MSCs.

In addition to this, we also did some further tests with different $w_{max}$ (please see the answer for the next question). A higher $w_{max}$ dose increase the IWC for the BASE-4ICE simulation, although the impacts are relatively small and will not change the conclusion that BASE-4ICE simulation significantly underestimated the IWC in both altitude ranges. Increasing from $w_{max}$=10 m s$^{-1}$ to 15 m s$^{-1}$ for CTR simulation, the IWC will increase 20 % and 50 % for 6-7 km and 11-12 km. For SIP-4ICE simulation, the increases will be 1 % and 63 % respectively. The increased values ($w_{max}$=15 m s$^{-1}$) of IWC of CTR simulation are still much lower than the non-increased value ($w_{max}$=10 m s$^{-1}$) of SIP-4ICE simulation. For higher altitude between 10 and 11 km, the increase of IWC for SIP-4ICE simulation will only improve the agreement between SIP-4ICE and observation. However, since the scope of the paper is to study the impact of adding SIP in the simulation, we think it's better to keep the $w_{max}$ the same for all simulations. A further clarification is added to the text to briefly describe the possible impact factor of $w_{max}$ (please see the answered of the next question).

Did you do any simulations with a stronger updraft forcing to investigate the sensitivity to the updraft forcing?

We made additional tests with different maximum updraft forcing speeds. A stronger initial updraft forcing speed produces slightly stronger convection therefore larger IWC (0-20% increase below 9 km, up to 50% between 10-11 km). The impact on $N_i$ is range from -44% to 63% below 11 km. However, both SIP simulations ($w_{max}$=10 or 15 m s$^{-1}$) still produce much higher IWC and $N_i$ in most of altitudes between 5 and 11 km comparing to those from both CTR simulations. One particular note is that between the altitudes of 10 and 11 km, SIP simulations with $w_{max}$=15 m s$^{-}$

[1]. does produce higher IWC (63% higher mean IWC). Therefore, the underestimation of IWC shown in Fig 9b might be partly caused by the uncertainty of the strength of the convection. We added a short description at the end of section 5.3:

*"Another possible explanation of the underestimation is the uncertainty of the strength of simulated convections. In this study, the maximal updraft nudging speed $w_{max}$=10 m s$^{-1}$ is used as the default value. Simulations with different $w_{max}$ are also tested. Using $w_{max}$=15 m s$^{-1}$ in SIP-4ICE simulation will produce 63% higher averaged IWC between 10 and 11 km than that produced by default SIP-4ICE simulation with $w_{max}$=10 m s$^{-1}$. For SIP-4ICE simulation, the impact of $w_{max}$ for lower altitudes between 6 and 7 km is negligible. The CTR simulation with $w_{max}$=15 m s$^{-1}$ produces higher IWC compared to the default CTR simulation with $w_{max}$=10 m s$^{-1}$ (20% and 50 % higher for altitudes range of 6-7 km and 10-11 km respectively). However, these values are still significantly smaller than those of the default SIP-4ICE simulation."*

**Specific**

In the introduction it may be worth mentioning the recent study by James et al. 2021, ACP. On the importance of SIP during drop – ice interactions.

The study of James et al. (2021) was referenced following the reviewer's comment.

"systematic studies of the effect of SIP on cloud microphysics with the help of cloud simulations have begun only in the last few years" I think this is not very accurate, there have been other previous studies.

We change the phrase as:

*"For the last few years, there are many new efforts on systematic studies of the effect of SIP on cloud microphysics with the help of cloud simulations"*

Page 3, lines 92-95. It is quite vague to say this paper uses "simplified parameterizations" here, with no extra detail. I would prefer to have this in the paper where it could also be justified. As written, I don't have much confidence in the modelling. It gave me a negative feeling, without knowing exactly what was done at this stage.

We changed the phrase into:

*"In the absence of a consensus on SIP parameterizations, these two processes were described by two specific parameterizations proposed in the literature, which provide a sufficient enhancement of Ni above the melting layer consistent with in situ observations in the MCSs"*

Equation 2: I am guessing that this is added to the vertical wind as an extra term? i.e. so wt = Equation 2 + w_model … otherwise there would be no downdraft. Is this the case? Also, Equation 2 only gives positive vertical winds.

Line 168 – typo on this line "there have been"

Done.

Equation 3 does not give any dependence on temperature, whereas we know there is a thermal peak around -15 deg C. I think this could be mentioned / clarified in the text. Could this affect your results? Maybe the thermal peak could lead to more ice in the 11-12 km bracket, if the ice particles were mainly formed higher up in the cloud, and did not have chance to grow to large sizes and precipitate out?

Very good question! In current version, the approach for FFD is not temperature dependent except for the temperature range. We are currently looking into the impact of the thermal peak around -15°C as well as the gradual activation of the FFD near the temperature of -3°C (particularly for winter temperature inversion cases, e.g. freezing rain). Both could have strong impacts on the simulation results under specific conditions. We are planning to deal with this topic separately with a dedicated study. On the other side, the current paper aims to demonstrate the impacts of just adding significant number of secondary ice in the high-res simulations while being aware that there are still large knowledge gaps on the SIP processes (very different FFD rates from literature and 5 other mechanisms are not modeled at all). We added a clarification in section 3.3.2:

*"Within the temperature range, the current parameterization is not temperature dependent. Further studies are undergoing for exploring the impacts of variation due to temperature."*

Page 16, point 2 line 486. There is an argument by Wojciech Grabowski (https://acp.copernicus.org/articles/21/13997/2021/acp-21-13997-2021.pdf) that the increase in temperature obtained is offset by the weight of condensate in the air (see argument on page 2 of the above paper). It seems to be in conflict with your point 2. Maybe it is the vapour growth rather than the freezing step that leads to the increase in buoyancy?

The series of works by Grabowski (including the one mentioned above) are related to the problem of convection invigoration in liquid clouds. The time of phase relaxation ($\tau_p$) in liquid clouds is typically quite low, and it ranges from a few seconds for maritime clouds to tenths of a second for continental convective clouds. The time of phase relaxation characterizes how quickly the system consisting of a population of cloud droplets and water vapor will get to equilibrium. In the consideration of invigoration of convection, it is important how much water vapor excess is available and how quickly the excess water vapor above the saturation point can be depleted by droplets to generate buoyancy sufficient for the convection invigoration. The main point of Grabowski's works is that in liquid convective clouds supersaturation and $\tau_p$ is typically small and enhancement of droplet concentration will result in a relatively minor increase of buoyancy. In our study, we are considering primarily ice clouds, which typically have a time of phase relaxation of the order of minutes and tens of minutes (e.g., Korolev and Mazin, 2003, JAS, https://doi.org/10.1175/1520-0469(2003)060<2957:SOWVIC>2.0.CO;2). Such cloud may remain

in high supersaturation over ice for a long time (e.g., Korolev and Isaac, 2006, JAS, Fig.10, https://doi.org/10.1175/AMSMONOGRAPHS-D-17-0001.1). In this case, if a large number of ice particles is introduced in such environment, they will rapidly deplete water vapor due to small phase relaxation time and release latent heat, which will induce and/or invigorate the convection.

In the point 2 of the conclusion, we stated that the latent heat from vapor deposition is the main source for the increase in buoyancy. However, one phrase in the section 6.2 appears confusing:

"With large RWC and rain mean-mass diameter, the rising parcels had a high SIP potential, which eventually led to increased freezing and increased latent heating and buoyancy, thereby enhancing secondary convection."

We change it to:

*"With large RWC and rain mean-mass diameter, the rising parcels had a high SIP potential, which eventually led to greater $N_i$, vapor growth on ice, increased latent heating and buoyancy, thereby enhancing secondary convection."*

**Reviewer 2 (Alain Protat):**

This paper investigates the importance of two SIP processes on tropical convection using high-resolution simulations. They also make use of observations from the HAI-HIWC program to provide some evaluation of the performance of the simulations.

I really enjoyed reading this paper, which is very well written and organized. I have a few comments, mostly regarding the comparisons with the observations, which would need to be addressed before the paper is ready for publication. These comments are listed below.

Thank you for your positive review and suggestions. We made many changes in the manuscript. The answers are in blue color.

**Main comments**

In the paragraph starting line 299, the authors comment that the differences in IWC statistics at 11-12 km could be due to differences in the sampling of data, with the HAIC-HIWC program targeting conditions with high IWC. However, the same comment applies to all comparisons in this paper, including those that seem to work like IWC at 6-7 km, and number concentrations. I think it is the main weakness of the paper not to have attempted to minimize these differences in sampling. There are different ways to address this problem. What I suggest would make more sense is to compare relationships between parameters. For instance, Ni versus reflectivity, IWC versus reflectivity, at different heights (6-7 and 11-12km like you've done is good) so that you can compare in different ranges of reflectivity how the Ni and IWC distributions compare with the observed ones. Or use vertical velocity like you've done later in the paper when looking at SIP rates. I am very confident that with a little more care you will be able to say more about how well the simulation captures what the observations say.

Thanks for the suggestion! This is potentially a very useful tool for diagnosing model deficiency. We added in the section 5.4 of the manuscript more results with 2D histograms of $N_i$/IWC and reflectivity (new figures 13 and 14). These new results point to the issue of overestimation of the reflectivity even from SIP-4ICE simulation which estimated quite well the total $N_i$ and IWC. We further looked at the individual ice categories with the same method which gave us indication of possible cause of the overestimation. Note that the simulated reflectivity from P3 scheme is not from an instrument simulator. The calculation in P3 integrates over the particle size distribution, taking into account considering the different mass-diameter relations in different size regimes, but it is based on Rayleigh scattering (no attenuation, Mie and multiple scattering, etc.). Therefore, one should be somewhat cautious when using the reflectivity from the model (and not overly quantitative), e.g. high model reflectivity (>25 dBZ) may not be exactly linked to similar conditions in the observed convection. With this in mind, we decided to keep the original format of presenting the results in Fig. 8 and 9, and use the 2D histograms method to diagnose the overestimation of reflectivity from the model simulation. Several possible reasons are given to

explain the overestimation of the reflectivity between 6 and 7 km (please see the response to the later question about the cause of high reflectivity in the simulation).

Results from Fig. 6, especially IWC and Z: This is a good illustration of my previous point, Differences in IWC and Z should result in very different IWC-Z relationships from the two simulations. You could compare simulations in IWC – Z space and compare with observations of IWC and Z (simulated from PSDs or measured by NAX close to the aircraft) from HAIC-HIWC too. Regions of simulated storms that were not measured by the aircraft will not be covered by observations, but that's OK, just compare where you can in the range of possible IWC and Z values.

Please refer to the previous answer.

Fig. 12: The Convair flew slightly above or slightly below the 0degC isotherm. If not corrected for attenuation, your stats from the Convair cannot be used as a reference. Has attenuation been corrected? I suspect not.

During the data sampling the Convair-580 always flew above the melting layer: typically, between 6 km and 7.5 km. We did not account for the X-band attenuation in the melting layer and the rain below, since this data were not used in statistics, and therefore, they are outside the scope of the paper. The focus of the paper is the ice region of MCSs formed above the melting layer. Even though some diagrams show comparisons below the melting layer, these results were not discussed in the paper and should be considered as complimentary. As shown in Matrosov (2008, IEEE, Fig.7, https://doi.org/10.1109/TGRS.2008.915757) the accounting for the attenuation in the melting layer is estimated as a few dB at most for the studied clouds. This will not affect the diagram in Fig.12.

A maximum of radar Z in liquid of ~25 dBZ with a maximum at ~ 40 dBZ at the right side of the CFAD seems too low for tropical convection at X-band.

The X-band radar CFAD in Fig.12 is a result of averaging over fourteen MCSs studied during the Cayenne campaign. Even though the X-band radar data set includes only cases with precipitation reaching the ground level (no precipitation cases were filtered out), the statistics contains many cases with weak precipitation below the melting point. This explains the bias of the average Z towards low values.

Comparisons from Fig. 12: I think there is more to say about this figure. I also think the colour scale of your reflectivity CFADs is not well chosen it does not highlight enough the maxima that are of most interest:

At the manuscript preparation stage authors spent some time playing with color map and scale for the CFAD in Fig.12. The authors came to a conclusion that the CFADs shown in Fig.12 is probably the best that could be achieved.

- just below the melting layer there is a local maximum of rainfall in the SIP simulation at ~ 25 dBZ, where the observations show a maximum. It corresponds to a blob of higher frequencies ion the CFAD produced by SIP around +15 dBZ above the melting layer. It is very impressive! Probably the most important result I see in those comparisons.

   Thanks for the comment!

- The SIP simulation still produces very large ice particles (30 dBZ+) in the HM region that are completely absent from the observations. That is a long-standing problem in tropical convection simulations, it needs to be mentioned and discussed. Used to be associated with way too much graupel produced by the models. But in your case you don't use the same type of parameterizations, so what causes this population of ice particles to be there ? Calls from a deeper analysis of your four ice categories ? Is there one in particular that occupies this space on the reflectivity CFAD ? How are the four ice categories spread out on the CFAD ? That would be a really interesting plot to make and analyze, maybe.

   Thanks for these questions! We added further investigation in the section 5.4. 2D histograms of $N_i$/IWC and reflectivity are used to investigate the cause of the overestimation. Some possible causes are explained in the section 5.4:

[revised manuscript text omitted]

Following up from the previous comment, nothing is done to show how the four ice categories end up looking like and how they contribute to the total Ni, IWC, etc …

Please see the answer for the previous question.

**Other comments:**

Line 168: " There have BEEN" ("been" is missing I think)

Done.

Line 177: Mossop (not Mossp)

Done.

Section 4, lines 239 - 240: it would be interesting to document somehow what difference this choice makes to the morphology of the storm for SIP simulations. Would it be possible to compare cross sections (horizontal or vertical) as well ? Or comment on how much that choice changes the morphology of the storm without showing figures? Seems like a good opportunity.

Good point! We check the morphology of the storm for 3 SIP simulations with different number of ice categories (Fig. D1). They look very similar to each other for the TOA LW fluxes. We found some differences for the reflectivity, e.g. the reflectivity values of SIP-2ICE between 5 and 10 km are generally slightly higher than those of the other two cases. This agrees very well with the results presented in Figure 5. Because the storm morphology does not change significantly with

different number of ice category, we prefer to only show the profiles in Fig. 5. We also added a short description in the manuscript to address this point:

*"Note that adding more ice categories for baseline and SIP simulations will not change significantly the morphology of the storm."*

[Figure]

**Figure D1. Longwave flux at the top of atmosphere (panel a-c) and simulated radar reflectivity (d-f) for the cross-sections indicated by the black lines in (a-c, respectively). SIP-2ICE: panel a and d, SIP-3ICE: b and e, SIP-4ICE: c and f. All data are from the 120 min time step after the model initiation.**

Figure 7: The only parameter that shows much less difference between simulations in the ensemble exercise is vertical velocity. Any idea why ? You should offer more explanations about that, and acknowledge that as well.

Unlike the other parameters which are either mean or median value for a given model level, the $W_{max}$ is single maximal value of vertical velocity for a model level. Without the averaging process, the profiles of $W_{max}$ can be noisy and with wider distributions as we see in the Fig. 7. The profiles of $W_{max}$ of CTR and SIP-4ICE overlap quite a lot. However, if we look at the mean profile of all ensemble members, there are still distinguishable difference above 6 km (solid lines). Also, the Fig. 7 only shows a single time step at 120 min. For the other time steps, the difference could be more significant such as at 90 min (Fig. D2). We added a short description in the manuscript:

*"The vertical velocity $W_{max}$ of an ensemble member is the single maximal value of a give model level. Therefore, the profiles of $W_{max}$ can be noisy. The ensemble profiles of $W_{max}$ of CTR and SIP-4ICE overlap, especially below the altitude of ~6 km. However, the averaged maximal vertical velocity $W_{max}$ (solid lines) diverges above the altitude of ~6 km, which indicates that the SIP simulations generate stronger updrafts in general."*

[Figure]

**Figure D2. Similar to Fig. 7 but for 90 min.**

Lines 285-288: Here you need to talk about the width of the distribution. Although the max in obs is incidentally close to the CTR simulation, the width of the distribution is much better captured by the SIP simulation. Then you need to investigate why there is this high population of 10^6 $N_i$ in the simulation, maybe with some sensitivity tests ? If it's good at 6-7 km, but not at 11-12 (higher concentrations), it seems unrelated to how well you simulate SIP maybe?

For the width of the distribution, we added a brief description in the text.

For the overestimate by SIP-4ICE around $3 \times 10^5$ m$^{-3}$, the answer is probably directly linked to one of the purposes of this paper. On one side, there are still large uncertainties with regard to the SIP parameterization (several mechanisms are not parameterized at all, and for FFD we still need better quantification, e.g. temperature dependency, etc.). On the other side, the ice-ice collection efficiency is largely uncertain with no consensus in the scientific literature. The adjustment between SIP (mostly FFD) and ice collection efficiency might produce the results we want, but might not reflect the reality (need more laboratory/modeling works to quantify SIP and collection efficiency).

That being said, we do think SIP-4ICE produce a better estimation of $N_i$ between 10-11 km as it produced a better distribution. In our future work, we need to refine the parameterization of FFD, e.g. better temperature dependency with a thermal peak at -15°C and gradual activation of FFD. These changes might change the production of secondary ice and therefore change the $N_i$ at higher altitude. However, this is out of the scope of the current paper. The main purpose of this paper will be to show that adding SIP in the NWP simulation is crucial to produce the high $N_i$ and high IWC observed in MSCs.

The modified paragraph is as follow:

*"The general behaviour of the functions $F(N_i)$ for $11 < H < 12$ km (Fig. 8b) is similar to that obtained for $6 < H < 7$ km in Fig.8a. The maxima of $F(N_i)$ for CTR and observed values correspond to approximately the same ice concentrations of $10^5$ $m^{-3}$. However, the maximum of $F(N_i)$ in CTR is nearly triple that of the observed $F(N_i)$ which was reasonably close to that of SIP-4ICE. The width of $F(N_i)$ of SIP-4ICE agrees better with the observation than that of CTR. There are almost no grid points in CTR with a concentration above ~$3{\times}10^5$ $m^{-3}$, whereas SIP-4ICE overestimated $F(N_i)$ compared to the measured values. The overestimation of $N_i$ around $3{\times}10^5$ $m^{-3}$ is probably caused by uncertainty in the parameterization of both FFD process and ice-ice collection efficiency. This warrant further studies on better quantifications of both processes."*

Section 5.4 : differences in LW radiation are really interesting and show substantial differences between non-SIP and SIP comparisons. Why didn't you attempt to compare with satellite products?

We had considered comparing satellite product with the simulation. However, the simulated storm is idealized one and thus it does not have the same size and complexity as the real tropical MSC system. Therefore, the direct comparison may not give us convincing results. We preferred to only compare between the two model simulations (CTR and SIP-4ICE) so that they are more comparable. For our future work with real case simulations, the satellite products will certainly be used to valid the simulation.

Lines 367-368: If I understand correctly panel a is not normalized by the actual areas of the three updraft categories. Maybe the lower values for downdrafts is just because there is much less area actually occupied by downdrafts? I wonder if it would be complementary to look at an activation fractional area (% of downdraft area where SIP is doing something)?

You are right, the actual fraction of active SIP is more pronounced in downdrafts than the two other categories. We added new subplots in Figure 17 to show the fraction of SIP area.

[Figure]

**Figure 17.** (a): area in m² with active HM process; (b): fraction of area with active HM process; (c): average rate of SIP by the HM process within the active area shown in (a); (d): total SIP rate, product of (a) and (c). (e) to (h): same as (a) to (d) but for the FFD process. Red lines (updrafts): SIP with $w > 3$ m s⁻¹, blue lines (outside of updrafts/downdrafts): SIP with $w$ between -3 and 3 m s⁻¹, yellow lines (downdrafts): SIP with $w < -3$ m s⁻¹. All results are temporal averages between 90 and 120 min.

Lines 392-395 and line 427: This reads like you conclude it's like that in real life tropical convection. Most of this is driven by the choices you made for the parameterization, so maybe this comment should be restricted to "in these simulations ...". The same caution should be exercised in the general conclusions, I think.

Agree. We made these changes:

*"The results obtained show that the vertical extent ΔH of FFD is deeper and its rate is higher than those of HM. This finding leads to the conclusion that in the simulations of this study the overall contribution of FFD in the production of the secondary ice in tropical MCSs is significantly higher than HM."*

*"As mentioned in subsection 6.1, in the simulations of this study, the FFD process plays a dominant role in SIP compared to the HM process."*

*And in the conclusion part:*

*"Analysis conducted in this study lead to the following general conclusions for the high resolution NWP simulations:"*

Line 451: We sort of discover here that your objective is to test whether a higher collection efficiency has large influence on the result. I think you should state clearly in lines 444-448 that you are taking an extreme case of higher efficiency to explore. This also calls for another question: why did you chose the default one you used? Is it known to be more physically realistic?

Agree. We added a description as follow:

*"As seen from Eq. (5) the new $e_{ii}$ varied linearly from 0.1 to 1.0 within the temperature range -20° < T < -5°C. For T < -20°C, $e_{ii}$ = 0.1, and for -5°C < T < 0°C $e_{ii}$ = 1.0. For all temperatures, $e_{ii}$ in (5) is much higher than that in (4). We want to use this high $e_{ii}$ parameterization to show the significant impact of $e_{ii}$ on the simulation results."*

For the choice of Cotton et al. (1986) as the default parameterization, due to the lack of consensus in the literature, we try to use one with mid-range efficiency. Again, this also point to the crucial need for a better quantification of the ice-ice collection efficiency. We also made this change:

*"There is a diversity of parameterizations of $e_{ii}$ employed in models (Khain and Pinsky, 2017), which may vary by up to three orders of magnitude. In this study, the mid-range $e_{ii}$ in (4) is used as the default parameterization. However, the uncertainty in $e_{ii}$ raises a question about the impact of the ice aggregation parameterization on the $N_i$ and high IWC formation."*

Fig. 20: it is striking that the vertical velocities are a lot lower at 1km compared to 250m. This should be mentioned I guess? Also I think it would be interesting to add the profiles of Fig. 6 (as dashed line) on Fig. 20 to support this discussion.

We found in our simulation that the simulated maximal vertical wind speed is inversely proportional to the horizontal grid spacing of the NWP model. In this topic, there are detailed discussions in Weisman et al. (1982), particularly in results presented in Fig. 17.

Reference: Weisman, M. L. and Klemp, J. B.: The dependence of numerically simulated convective storms on vertical wind shear and buoyancy, Mon. Weather Rev., 110, 504–520, 1982.